

# A laboratory investigation of the ice nucleation efficiency of three types of mineral and soil dust

Mikhail Paramonov[1], Robert O. David[1], Ruben Kretzschmar[2], Zamin A. Kanji[1]

[1]Institute for Atmospheric and Climate Science, ETH Zürich, Switzerland
[2]Institute of Biogeochemistry and Pollutant Dynamics, ETH Zürich, Switzerland

*Correspondence to*: Mikhail Paramonov (mikhail.paramonov@env.ethz.ch) and Zamin A. Kanji (zamin.kanji@env.ethz.ch)

**Abstract.** Surface-collected dust from three different locations around the world was examined with respect to its ice nucleation activity (INA) with the Portable Ice Nucleation Chamber (PINC). Ice nucleation experiments were conducted with particles of 200 and 400 nm in diameter in the temperature range of 233–243 K in both deposition nucleation and
condensation freezing regimes. Several treatments were performed in order to investigate the effect of mineralogical composition, as well as the presence of biological and proteinaceous, organic and soluble compounds on the INA of mineral and soil dust. The INA of untreated dust particles correlated well with the total feldspar and K-feldspar content, corroborating previously published results. The removal of heat-sensitive proteinaceous and organic components from the particle surface with heat decreased the INA of dusts. However, the decrease in the INA was not proportional to the amount
of these organic components, indicating that different proteinaceous and organic species have different ice nucleation activities, and the exact speciation is required in order to determine why dusts respond differently to the heating process. The INA of certain dusts increased after the removal of soluble material from the particle surface, demonstrating the low INA of the soluble compounds and/or the exposition of the underlying active sites. Similar to the proteinaceous organic compounds, soluble compounds seem to have different effects on the INA of surface-collected dusts, and a general conclusion about how
the presence of soluble material on the particle surface affects its INA is not possible. The investigation of the heated and washed dusts revealed that mineralogy alone is not able to fully explain the observed INA of surface-collected dusts at the examined temperature and relative humidity conditions. The results showed that it is not possible to predict the INA of surface-collected soil dust based on the presence and amount of certain minerals or any particular class of compounds, such as soluble or proteinaceous/organic. Instead, at temperatures of 238–243 K the ice nucleation activity of the untreated,
surface-collected soil dust in condensation freezing mode can be roughly approximated by one of the existing surrogates for atmospheric mineral dust, such as illite NX. Uncertainties associated with mechanical damage and possible changes to the mineralogy during treatments, as well as with the BET surface area and its immediate impact on the number of active sites $n_{s,BET}$ parameterisation, are addressed.





# 1    Introduction

Atmospheric aerosol particles are well-known to modify the microphysical properties of clouds, such as their albedo, lifetime and precipitation patterns in what is known as the indirect effects of aerosols on climate (e.g. Lohmann and Feichter, 2005). Due to the importance of clouds in the Earth's hydrological cycle and energy and radiation balance, these aerosol-induced changes are a subject of rigorous research. However, due to the complexity of aerosol-cloud-climate interactions, the exact quantification of the radiative forcing associated with changing cloud properties has been challenging (Boucher et al., 2013).

In the atmosphere aerosol particles can form both cloud droplets and ice crystals; in such cases these aerosol particles are referred to as cloud condensation nuclei (CCN) and ice nucleating particles (INP) forming liquid and solid hydrometeors, respectively. While the studies of cloud droplet formation have been numerous (e.g. Andreae, 2009 and references therein; Kerminen et al., 2012; Paramonov et al., 2013), the studies of atmospheric ice nucleation have been insufficient and still pose a lot of open questions (DeMott et al., 2011; Kanji et al., 2017). In general, a significant fraction of precipitating clouds in various environments contains ice, e.g. both in the tropics (Lau and Wu, 2003) and in the Nordic countries (Sporre et al., 2014). Primary nucleation of ice in the atmosphere can occur in the absence of insoluble material, by way of freezing of supercooled liquid water droplets – this process is known as homogeneous freezing, and it requires temperatures below ~ −37 °C (236 K) (e.g. Murray et al., 2010; Vali et al., 2015). However, the presence of INP results in the onset of freezing at warmer temperatures in what is known as heterogeneous freezing. This process has four known mechanisms: deposition nucleation, condensation freezing, immersion freezing and contact freezing (Vali, 1985; Vali et al., 2015). Condensation and immersion freezing are thought to be the dominant pathways for the first ice formation in mixed-phase clouds (de Boer et al., 2011). Since the concentration of ice crystals in the atmosphere is typically orders of magnitude larger than the INP concentration (Cantrell and Heymsfield, 2005), several secondary production mechanisms are also believed to contribute to the observed ice crystal number concentrations (e.g. Hallett and Mossop, 1974; Field et al., 2017).

Several aerosol species of atmospheric relevance have been studied and shown to act as efficient INP. Biological aerosols (bioaerosols) have long been postulated to act as INP (e.g. Maki et al., 1974; Lee et al., 1995; Möhler et al., 2007; Delort et al., 2010). Even though well-known for the exceptionally low affinity for water (Pringle et al., 2010), black carbon (BC or soot) has also been studied as potential INP (e.g. DeMott, 1990; Gorbunov et al., 2001; Dymarska et al., 2006; Kanji and Abbatt, 2006). More recently INP of marine origin, those associated with marine biological activity and sea spray production, have seen a resurgence in interest as well (e.g. Després et al., 2012; Burrows et al., 2013; Wilson et al., 2015; DeMott et al., 2016; Ladino et al., 2016; McCluskey et al., 2017). Perhaps the most well-known INP species, primarily so due to its size, morphology, significant presence in the atmosphere and the ability to be transported great distances, is mineral dust (e.g. Kumai, 1961; Cantrell and Heymsfield, 2005 and references therein; Kellogg and Griffin, 2006). Originating predominantly from major deserts, such as the Sahara and Gobi, mineral dust particles are easily lifted from soils





with low vegetation cover by surface winds, with annual emission rates on the order of several thousand Tg per year (Engelstaedter et al., 2006). Due to its sources, the occurrence and the amount of mineral dust in the atmosphere varies greatly both spatially and temporally (Tegen and Fung, 1994). Many studies have concentrated on the ability of mineral dust to form ice at the atmospherically relevant thermodynamic conditions (e.g. Hoose and Möhler, 2012; Murray et al., 2012;

Kanji et al., 2017 and references therein), and it has been reported as the most common INP in the formation of cirrus clouds (DeMott et al., 2003; Cziczo et al., 2013). Mineral dust has a size-dependent efficiency with respect to its ice nucleating activity (INA), with larger particles activating at higher temperatures and lower supersaturations (Archuleta et al., 2005). However, the exact mechanism of ice nucleation on mineral dust is not fully understood, although it is strongly dependent on its mineralogical composition (Pinti et al., 2012), morphology (Baustian et al., 2012) and coating (Möhler et al., 2008). This

complicates the representativeness of ice formation on mineral dust in global models as both the concentration of mineral dust INP and their mineralogical composition vary greatly depending on location. Atmospheric processing and aging during the transport of mineral dust (Kanji et al., 2013), as well as the effect of coating, are also not fully understood (Sullivan et al., 2010a).

In the atmosphere a typical mineral dust particle is composed mainly of clay minerals and quartz, especially if found close to

the emission source (Broadley et al., 2012; Murray et al., 2012). However, during long-range transport aerosol particles partake in various photochemical, oxidative and aging processes, and both organic matter and biological material may condense/adsorb onto the surface of the particle, potentially altering its size, surface properties and, possibly, INA (Krueger et al., 2004; Hinz et al., 2005; Dall'Osto et al., 2010). Besides the mineral dust originating from major deserts, dust of agricultural origin is also of atmospheric importance, signified by the estimated 20-25% contribution to the global

atmospheric dust load (Ginoux et al., 2012; Boucher et al., 2013). The chemical composition of arable soil dust is frequently more diverse than that of naturally emitted desert dust (Simoneit et al., 2004), illustrating its more complex role in atmospheric ice nucleation (IN) processes.

In recent years, surface-collected mineral dust of both natural sources and those influenced by anthropogenic activities has been investigated with respect to its INA. Field et al. (2006) examined the INA of two types of desert dust, from the Sahara

and Asia, and found no significant difference between their INA. The study reported that at temperatures above 233 K no significant ice nucleation via deposition nucleation occurred, with droplets forming simultaneously with ice crystals via the condensation freezing mode, possibly due to the presence of some soluble material on the surface of the particles. The maximum activated fraction (AF) at 253 K was ~ 5-10%, increasing to 20-40% at temperatures below 233 K. Kulkarni and Dobbie (2010) also studied several types of Saharan dust, as well as one Spanish dust, and demonstrated that the onset of ice

nucleation occurred at relative humidities with respect to ice ($RH_i$) as low as 104%. The small differences in the observed INA between Saharan and Spanish dusts was attributed to the increased content of Ca-rich minerals in the Spanish dust (e.g. calcite) and, therefore, its lower INA. A comprehensive study by Boose et al. (2016a) determined the INA of several types of





both airborne and surface-collected dust in the immersion freezing mode in the temperature range of 233−263 K. The mineralogical analysis indicated that at temperatures above 250 K the INA is driven mainly by the feldspar content of the dust, while at lower temperatures the sum of feldspar and quartz becomes more important. Kaufmann et al. (2016) also explored the INA of several types of surface-collected dust in the immersion mode. They found that mineralogy was only

able to fully explain the observed INA of five out of twelve dust samples; for the remainder of the dust samples mineralogy was only able to explain part of the observed INA or played no role at all. The study also reported that microcline was the most efficient ice nucleating mineral, and that while mineralogy is an important aspect in determining the INA of mineral dust, the mixture of minerals in a dust particle results in similar IN capabilities of different dusts regardless of their exact composition.

Both dry heating and the treatment of dust with hydrogen peroxide ($H_2O_2$) are known to denature the biological proteinaceous matter (Pouleur et al., 1992) and digest all organic matter (Conen et al., 2011), respectively. Several recent studies have used these techniques in an attempt to decipher the effects of biogenic dust components on the INA of soil dust. Conen et al. (2011) reported that biological residues are mainly responsible for the INA of surface-collected soil dust in the immersion mode. The study found that in the temperature range of 258–269 K the INA of soil dust particles was reduced

significantly after heating to ∼ 373 K, demonstrating the importance of heat-sensitive compounds likely of organic origin. O'Sullivan et al. (2014) collected fertile soil dust from several locations in England and examined its INA in the immersion mode, reporting that at temperatures above 258 K the INA of soil dust is diminished after heat treatment and even more so after $H_2O_2$ treatment, pointing to the removal of biological matter (Pouleur et al., 1992) and its importance for the INA in this temperature regime. At temperatures below 249 K, the INA of soil dust, as measured by the number of active sites $n_s$,

did not significantly differ between untreated samples and those heated to 363 K, suggesting that inorganic components dominate the ice active sites in this temperature regime. Similarly, Tobo et al. (2014) investigated the INA of arable soil dust from Wyoming, USA in the water supersaturated regime above 237 K. The study reported that both $H_2O_2$ treatment and heating to 573 K reduced the INA of the soil dust in a similar manner, with a distinct temperature dependence. At 237 K the difference between the untreated and heated/$H_2O_2$-treated dust was minimal, if present at all; however, this difference

gradually increased as the temperature increased. The two aforementioned studies both agreed that as the temperature decreases towards the homogeneous freezing temperature, the importance of biogenic constituents in the INP population decreases in favour of inorganic components. Hill et al. (2016) also confirmed that the removal of soil organic matter (SOM) from soil dusts either by heat or $H_2O_2$ significantly reduced the number of active INP at temperatures warmer than 255 K.

It is clear that the interactions of mineral dust with water vapour in the atmosphere under the cirrus and mixed-phase cloud

regimes is a complex phenomenon. The motivation for this work is to disentangle the effects of the multiple mineral dust constituents on its ice nucleation activity. This work aims to establish the effect of size, surface area, soluble material, biological proteinaceous matter and total organic matter on the INA of surface-collected mineral and soil dust. This is



achieved by a comprehensive analysis of three surface-collected dust samples, representative of both natural and anthropogenic origins. In addition to IN measurements, several techniques are used to characterise the dust samples and aid in the interpretation of the IN measurements. The goal is to provide an overview of mineral dust components responsible for the INA in deposition nucleation and condensation freezing modes in the temperature range of 233−248 K.

## 2 Methodology

### 2.1 Investigated dust samples

Three dust samples were investigated for their INA in this work. These dusts are natural, surface-collected dusts from Iceland, China and the Himalayas. The dust from Iceland is a glaciogenic silt collected from the glacial river Mulakvisl about 10 km from the Myrdalsjokull glacier in Southern Iceland; the material originates from the Katla volcanic system

under the glacier. The dust from China was collected at a remote location 100 km north-west of Hohhot. The location is a sparsely populated, dry and windy agricultural area next to a small rural road, with freely roaming cattle. The dust from the Himalayas was collected from the top of a glacier in the Khumbu Valley, about a 10 min walk from the Pyramid Station. Hereafter, the dusts are referred to by their origin, i.e. Iceland, China and Himalaya. To allow for ice nucleation experiments, the collected dust samples were dry sieved to select only the particles below 45 μm in diameter.

Besides the untreated dust, the samples were subjected to three treatments. The motivation behind conducting the three dust treatments was to remove certain compounds from the particle surface, to consequently examine the INA of the treated dust and to compare the results to other previously published studies. In order to remove biological proteinaceous matter, the dust samples were heated to 573 K for two hours (Pouleur et al., 1992). The procedure was carried out in a drying oven, and the samples were cooled at room temperature following the heating.

In order to remove the soluble material, the dust samples were washed using the following technique (Welti, 2012). Ten grams of dust were suspended in 300 ml of Milli-Q water, and the suspensions were first agitated for 15 minutes with a magnetic stirrer and then sonicated in an ultrasonic bath for 100 minutes. The suspensions were then centrifuged for 15 minutes to separate the dust from the supernatant water, which was retained for further analysis. The washing cycle was repeated three times for each dust, and after the last centrifuging procedure, the washed dust was dried overnight in an oven

at a temperature of 303 K.

To digest all organic matter, the dust samples were treated with $H_2O_2$ (O'Sullivan et al., 2014; Tobo et al., 2014) by adding 50 ml of 35% $H_2O_2$ solution to 10 grams of dust. Once the bubbling subsided, the suspensions were heated up to 323–333 K. Additional $H_2O_2$ was added to the heated suspensions until the bubbling completely stopped. The suspensions were kept overnight and then diluted by a factor of 100 with deionised water. After the dilution, the suspensions were centrifuged, and





the treated dust was then dried overnight in an oven at a temperature of 303 K. In the following discussion the treated dusts are referred to as heated, washed and $H_2O_2$-treated.

The mineralogical composition of the dusts was examined using a powder X-ray diffraction technique with subsequent Rietveld quantitative phase analysis. The samples were finely powdered using a McCrone micronising mill (Retsch, Haan, Germany) and packed with random orientation into rotary disk sample holders. X-ray diffractograms were recorded in Bragg-Brentano geometry on a D8 ADVANCE diffractometer using Cu Kα radiation (λ=1.5406 Å; 40 kV/40 mA) and a high-resolution energy-dispersive 1-D detector (LYNXEYE XE, Bruker AXS GmbH, Karlsruhe, Germany). Diffractograms were recorded from 4 to 70 °2θ with a step size of 0.015 °2θ and 10 s acquisition time per step. To quantify the amounts of X-ray amorphous material in the dusts, they were additionally analysed after adding 50 wt% of well-crystalline corundum powder as internal standard (CAS: 1344-28-1; purum p.a., Fluka). Following possible phase identification supported by the PDF-2 database (International Centre for Diffraction Data), Rietveld quantitative phase analysis was performed using TOPAS of the Bruker DIFFRAC.SUITE software package (Version 5).

In addition to the IN measurements described in the next section, several auxiliary bulk measurements were performed to characterise the dust to aid the interpretation of the IN results. In order to investigate the presence of heat-sensitive biological proteinaceous matter, thermogravimetric analysis (TGA; Perkin Elmer Pyris 1) was performed. The procedure involved the heating of a known amount of the dust sample to the desired temperature at a desired rate while the mass of the sample was closely monitored and recorded. The three untreated dust samples were heated from 303 K to 573 K at a rate of 10 K min$^{-1}$, kept at 573 K for one hour, then heated further up to 623 K at the same rate and kept at that temperature for 20 minutes. For better statistics and reproducibility, TGA was repeated on each untreated dust sample three times. Total carbon content in the untreated and heated dust samples was determined using a CHNS analyser (CHNS-932, LECO). The specific surface area of all untreated and treated dust samples was determined using the BET (Brunauer et al., 1938) 11-point adsorption/desorption isotherms measurements performed with the Autosorb-1MP surface area analyser (Quantachrome, Germany). The relative pressure range p/p$_0$ during the measurement was 0.05−0.3, and the gas used was $N_2$. The samples (approximately 1 g) were dried in a vacuum at 353 K for 15 hours prior to the analysis. The BET surface area is expressed in m$^2$ g$^{-1}$. Due to the amount of time required for the each measurement, the BET surface area of each untreated and treated dust sample was measured only once. To investigate the presence of soluble material in the untreated dust samples, the supernatant water after the three washing procedures was examined with respect to its pH and conductivity. pH was measured using a standard pH meter (Metrohm, 691 pH Meter) which is calibrated daily and has a calibration slope of > 0.97. The conductivity was measured using a conductivity probe (WTW Multi 350i Universal Pocket Meter). For the Scanning Electron Microscopy (SEM) images the untreated dust samples were sputter-coated with 4 nm of gold (SCD50 Bal-Tec). InLens images were recorded at 5 kV with a Leo 1530 FEG SEM with Gemini column (Zeiss, Oberkochen, Germany).



## 2.2    Instrumentation

The dry dust samples were mixed with 100 μm bronze beads and aerosolised with a fluidised bed aerosol generator (TSI 3400A). Following aerosolisation, the sample flow passed through a cyclone to remove particles larger than 2.5 μm in diameter. After the cyclone the flow was split into two parts, with one being the exhaust line with a filter for excess flow and

the other being the sample flow of approximately 2 lpm. The sample flow passed through an impactor with an aerodynamic cut-off size $D_{50}$ of 900 nm (size at which 50% of particles are lost) and a molecular sieve dryer to reduce the relative humidity in the sample flow. The sample flow further passed through a radioactive source ($^{210}$Po) to neutralise the aerosol particles and achieve an equilibrium charge distribution. Aerosol particles with a known charge distribution then entered a Differential Mobility Analyser (DMA) column (TSI 3081), where the particles of a certain size were selected based on their

electrical mobility (Aalto, 2004). Once the pseudo-monodisperse particle distribution was achieved, the sample flow was split in two parts, with 1 lpm fed to the condensation particle counter (CPC, TSI 3010) to determine a total number of particles of a certain size. The other 1 lpm flow passed through another impactor with a $D_{50}$ of 900 nm and was further led to the Portable Ice Nucleation Chamber (PINC).

PINC is an instrument for ice nucleation measurements based on the principle of the Continuous Flow Diffusion Chamber

(CFDC; Rogers, 1988), and it has proven to be a useful and reliable instrument during both field (Chou et al., 2011; Boose et al., 2016b) and laboratory (Chou et al., 2013; Kanji et al., 2013; Burkert-Kohn et al., 2017) studies. It consists of two parallel ice-coated walls, between which certain temperatures and levels of supersaturation can be reached. When both walls are at the same temperature, the air between the walls is at 100% $RH_i$ and subsaturated with respect to water ($RH_w < 100\%$). As the temperature of the walls is changed, i.e. the temperature gradient between the walls is increased, higher levels of

supersaturation can be reached, up to $RH_w \geq 100\%$. As such, it is possible to scan the $RH_w$ in both sub- and supersaturated regimes at a constant temperature (the so-called RH scan). Therefore, measurements conducted with PINC are representative of both deposition nucleation ($RH_w < 100\%$) and condensation freezing ($RH_w > 100\%$) regimes. The aerosol sample flow of approximately 1 lpm is guided through the chamber, sandwiched between the sheath air flows of approximately 4.5 lpm each. As the particles travel along a nearly laminar sample flow, and depending on the RH and temperature inside the

chamber, a fraction of them may activate as CCN and INP, forming liquid droplets and ice crystals, respectively. The estimated residence time (i.e. time for activation and growth) of aerosol particles inside the chamber is nominally seven seconds. As the formed hydrometeors move further down in the chamber, they pass through a so-called evaporation section. In this section both walls are held at the same temperature, i.e. the air is saturated with respect to ice and subsaturated with respect to water. Under these conditions, liquid droplets evaporate and the ice crystals remain. At a high enough RH within

the chamber, the droplets grow large enough not to evaporate completely in the evaporation section; this is referred to as droplet survival and is typically defined as $RH_w$ at a certain temperature. This RH also delineates the maximum $RH_w$ for a given temperature at which PINC measurements are performed. After the evaporation section, the sample flow, containing



presumably only ice crystals and unactivated aerosol particles, passes through an Optical Particle Counter (OPC, Lighthouse 5104 Remote), which counts the total number of particles in the size range of 0.5−25 µm binning them based on their optical size. Even after the evaporation section there is still a possibility of small droplets surviving; therefore, based on the OPC measurements, a lower size limit of ~ 2 µm is set, and everything above this size is counted as ice crystals. The number of

ice crystals is assumed to be representative of the number concentration of the INP at a certain RH and temperature. Instrument characterisation and previous studies with PINC have shown that the temperature uncertainty inside the laminar flow is ±0.4 K, which translates to the $RH_w$ uncertainty of ±2% (Chou et al., 2011).

### 2.3    Experimental information and calculations

For the untreated dust samples, particle sizes of 100, 200 and 400 nm were investigated, and RH scans were performed at

four different temperatures: 233, 238, 243 and 248 K. As mentioned previously, measurements have been conducted in both sub- and supersaturated regimes with respect to water. After the initial batch of experiments with untreated dusts, no significant heterogeneous INA was found for particles of 100 nm at any of the temperatures and for the temperature of 248 K for any of the particle sizes. Thereafter, for the treated dust samples, measurements were conducted only with 200 and 400 nm particles at 233, 238 and 243 K. The discussion in this work omits the particle size of 100 nm and a temperature of 248 K

altogether. For each dust, for each size and for each temperature, the experiment was repeated three times to ensure reproducibility. A total of 82 experiments and 274 RH scans with PINC were performed, totalling almost 400 instrumental hours.

Activated fraction (AF) was calculated as a ratio of the number of INP as counted by the OPC downstream of PINC to the total number of particles entering the IN system as measured by the CPC upstream of PINC. The number of active sites ($n_s$)

density was calculated in order to normalise the AF by surface area and to make the measurements at different sizes comparable. The $n_s$ is a deterministic concept (Langham and Mason, 1958) based on the assumption that particles of different sizes exhibit uniform chemical composition. This parameterisation assumes that a particle surface has sites (e.g. steps, cracks or chemical functional groups) where ice nucleation takes place, i.e. where ice embryos form (Vali, 1966), and that these sites are uniformly distributed on the particle surface. The $n_s$ was calculated as follows:

$$n_s = \frac{-\ln(1-AF)}{SA},$$    (1)

where $n_s$ is the number of active sites in $m^{-2}$, AF is the dimensionless activated fraction, and SA is the surface area in $m^2$. The $n_s$ was calculated utilising a particle density of $2.6 \times 10^6$ g $m^{-3}$ and a shape factor of 1.3 (Hinds, 1999). The method of determining $n_s$ is based on the measurements of the BET surface area, and is hereafter denoted as $n_{s,BET}$. The $n_s$ can also be calculated using the surface area of a presumably spherical particle instead of its BET surface area, denoted $n_{s,GEO}$. Since the





surface area of a spherical particle is, by definition, smaller than any deviation from perfect sphericity, $n_{s,GEO}$ values are larger than $n_{s,BET}$.

## 3    Results and Discussion

### 3.1    Untreated dust

Figure 1 presents the values of the onset of ice nucleation (defined as AF = 0.001) for the three examined dusts for both sizes. The onset values are shown as a function of temperature and supersaturation with respect to ice ($S_i$). Several initial conclusions can be drawn from this figure. First, 400 nm particles are more active than 200 nm ones since they require a lower RH for the onset of ice nucleation. Such size dependence is expected as postulated by classical nucleation theory and reported in Archuleta et al. (2005). It is important to note, however, that the overall increase in INA between 200 and 400 nm

particles is rather small. In absolute AF terms, the increase is on average less than 0.01, reaching 0.02 at the highest measured $RH_w$ (not shown). Second, as expected, when temperature decreases, a lower RH is required to nucleate ice and, therefore, more particles are able to act as INP. Lastly, for almost all dusts and temperatures the onset occurs in the deposition nucleation regime, i.e. below water saturation. The only exception here is the Iceland dust at the warmest temperature of 243 K, where $RH_w > 100\%$ was required to observe the onset of ice nucleation for both particle sizes. This

indicates that Iceland dust is the least IN-active dust as is further discussed below. Also visible in Fig. 1 is that China dust is the most IN-active of the examined dusts as it requires the lowest $RH_w$ to observe the ice nucleation onset. Comparing the values seen in Fig. 1 to those published in a study by Kanji et al. (2011) reveals that at 243 K the ice nucleation onset conditions of the three examined dusts are similar to those of Saharan and Canary Island dusts. At 238 K the Iceland, China and Himalaya dusts are all more active with the exception of 200 nm Iceland dust particles, the activity of which is similar to

that of Saharan and Canary Island dusts. At 233 K all examined dusts and particle sizes are more IN-active than the Canary Island dust (Kanji et al., 2011). The referenced study examined polydisperse particles with a mode size of 200-300 nm in diameter. The values presented in Fig. 1 also compare well to those of Asian desert dust AD2 at 233 K, Saharan desert dust SD2 at 238 K, and Saharan desert dust SD19 and Asian desert dust AD1 at 243 K as presented in Ullrich et al. (2017). The study investigated particles in polydisperse mode below 5 μm in diameter and reported that the ice nucleation onset RH

increased with decreasing temperature.

It is important to note that, in absolute terms, the AF values for untreated dusts examined in this study are fairly low, even at the highest attainable RH. The highest possible $RH_w$ in PINC is limited by the droplet survival at 238 and 243 K (113 and 110%, respectively) and by the lowest attainable wall temperature at 233 K (101%). At 243 K the largest maximum AF observed for China particles of 400 nm in diameter barely reaches 0.01 (1%); at 238 K this value increases to 0.1 (10%).

Even at the coldest measured temperature of 233 K, a maximum of one-third of 400 nm China dust particles acted as INP. A



likely explanation for this is the deviation of dust particles from the laminar flow inside PINC. If particles deviate from the laminar flow, they are subjected to an $RH_w$ below that intended, thus leading to an underestimation of the INP concentration – an issue raised by DeMott et al. (2015) and recently discussed in detail by Garimella et al. (2017). The problem of particles deviating from the laminar flow is further addressed throughout the paper.

As mentioned previously, and if the AF values across the whole $RH_w$ spectra are examined (not shown), it is clear that amongst the three dusts China dust is the most active for all sizes and temperatures investigated. At 238 and 243 K, China dust is followed by Himalaya dust, with Iceland dust exhibiting the lowest INA. At 233 K Iceland dust is slightly more active than Himalaya dust. Figure 2 shows $n_{s,BET}$ values as a function of $RH_w$ for each untreated dust, size and temperature. Each data point is an average of $n_{s,BET}$ values binned in 1% $RH_w$ intervals. Figure 2 shows that, contrary to INA based on the

AF values, Himalaya dust is the most active of the examined dusts for all sizes and temperatures, followed by China and Iceland dusts. The BET surface area of the Himalaya dust was found to be smaller than that of the Iceland and China dusts, resulting in higher $n_{s,BET}$. There are some instances where the differences between dusts are not well pronounced (e.g. 200 nm Himalaya and China dust particles at 233 K in the deposition nucleation regime). However, to aid the interpretation of untreated dust results, the highest INA of Himalaya dust followed by China and Iceland dusts is assumed hereafter.

Besides the differences among dusts mentioned above, several other important conclusions can be drawn from Fig. 2. First, for most of the examined dusts there is a clear increase in INA from sub- to supersaturated conditions with respect to water, i.e. from deposition nucleation to condensation freezing regime, and this increase occurs at an $RH_w$ slightly below that of 100%. This is likely due to either the formation of liquid water on the particle surface at $RH_w$ close to 100% (Sullivan et al., 2010b) or the wall temperatures and, hence, $RH_w$ uncertainty in the particle laminar flow as discussed in the methodology

section. Second, not directly seen in Fig. 2, the $n_s$ of the 200 nm particles is larger than that of the 400 nm particles. The difference in $n_{s,BET}$ values between the two examined sizes suggests that 200 nm particles are more ice-active than 400 nm ones. While it is not possible to directly determine the reasons behind this observed difference, it may be possible that large particles contain more soluble material blocking the active sites and/or that small particles may contain more IN-active material on their surface, e.g. bacteria or active minerals. Particles of approximately 200 nm in size, including mineral dust

species, have previously been reported as constituting the majority of the INP found in the ice crystal residual size distributions (Mertes et al., 2007). The section about the dust treatments discusses in detail the presence of organic and soluble material in the dust samples, although without the differentiation based on the particle size.

Several previously published studies have compiled an overview of $n_s$ values and parameterisations for various types of atmospherically relevant INP species (e.g. Murray et al., 2012; Boose et al., 2016a; Ullrich et al., 2017). However, a

significant fraction of these studies reported $n_{s,GEO}$ values which are not directly comparable to $n_{s,BET}$ (Murray et al., 2012; Hiranuma et al., 2015). The $n_{s,BET}$ values in condensation freezing mode ($RH_w$ > 100%) from this study may be used to investigate how well the INA of examined dusts compares to that of other INP species in immersion mode. Both Broadley et



al. (2012) and Hiranuma et al. (2015) examined illite NX as a proxy for atmospheric mineral dust and reported $n_{s,BET}$ values that are within 1 order of magnitude of the $n_{s,BET}$ values of all three examined dusts investigated in this study at both 238 and 243 K. The $n_{s,BET}$ values of the three examined dusts are also similar to those of kaolinite presented by Murray et al. (2011). In the relevant temperature regime all three investigated dusts are more IN-active than pure quartz (Zolles et al. 2015) and

less IN-active than pure K-feldspar and Na/Ca-feldspar (Atkinson et al., 2013). At 238 K the INA of Iceland and China dust is also similar to that of montmorillonite (Atkinson et al., 2013). This comparison of $n_{s,BET}$ values reveals that the INA of surface-collected dusts at 238–243 K in condensation freezing mode can be roughly approximated by one of the surrogates for atmospheric mineral dust, such as illite NX, for example. It is important to note here, however, that the comparison of measurements in the immersion freezing and condensation freezing modes is not straightforward. It has been reported that

instruments of the CFDC-type, which are typically assumed to measure in deposition nucleation and condensation freezing modes, can experience a deviation of aerosol particles outside of the intended lamina and a subsequent underestimation of the INP concentration (DeMott et al., 2015; Garimella et al., 2017). In such cases, particles outside the lamina experience an RH below that intended/assumed and, therefore, these particles may not activate, reducing the INP concentration. Correction factors of 3 (DeMott et al., 2015) or 1.4−9.5 (Garimella et al., 2017) have been proposed to correct the INP concentration for

the particle lamina deviations, possibly allowing for a more accurate comparison of condensation freezing and immersion freezing mode measurements.

### 3.1.1    The role of mineralogy

Mineralogical composition has been shown to play a crucial role in explaining the ice nucleation behaviour of atmospherically relevant mineral dust (Hoose et al., 2008; Atkinson et al., 2013; Harrison et al., 2016). Table 1 presents the

mineralogy of the three investigated untreated dust samples. Since mineralogical analysis was performed in bulk and since $n_{s,BET}$ is used in the upcoming discussion to describe the INA of dusts, it is assumed that the bulk mineralogy is representative of the 200 and 400 nm particle surface mineralogy. This assumption is not very accurate as it is well-known that mineralogical composition is size-dependent (d'Almeida and Schütz, 1983; Knippertz and Stuut, 2014). However, within the scope of this work, it was not possible to conduct the mineralogical analysis of size-selected dust samples. Table 1

shows that a large weight percentage of each dust sample is made up of amorphous, non-crystalline material, with Iceland dust containing as much as ~ 65%. While the exact identification and speciation of this amorphous matter is not possible, it can be composed of biological components (Formenti et al., 2008), various carbonaceous organic compounds (Gómez et al., 2005; Deboudt et al., 2010), alumosilicates (Archuleta et al., 2005), non-crystalline iron minerals (Takahashi et al., 2011), soluble material (Zhu et al., 1997) and others. A more detailed investigation into the possible identity of the amorphous

matter and its effect on the INA occurs in the following section when discussing the results of the auxiliary measurements of





the untreated dusts. In this section an attempt is made to explain the observed INA based solely on the mineralogical composition.

Immediately obvious in Table 1 is the highest amount of total feldspar and K-feldspar in the Himalaya dust. Total feldspar contributes almost one-third to the mass of the bulk Himalaya dust. Keeping in mind that both Atkinson et al. (2013) and Harrison et al. (2016), among others, reported the highest INA for feldspar and specifically for K-feldspar, its content in the Himalaya dust correlates very well with its highest INA among the examined dusts as seen in Fig. 2. China dust is the second most active dust, and this is also supported well by its second highest total feldspar content among the dusts tested (~ 27%). Iceland dust contains the least amount of feldspar (~ 14%) and exhibits the lowest INA based on $n_{s,BET}$ values. Microcline has been reported to be the most active of the K-feldspars (Augustin-Bauditz et al., 2014; Kaufmann et al., 2016; Kiselev et al., 2016). It is present in China and Himalaya dusts, albeit in small amounts (< 5%). It is, therefore, not possible to draw any conclusions about the effect of microcline on the observed INA of the examined dusts. Plagioclase feldspar is also present in all three examined dusts in significant amounts, although less than K-feldspar. Plagioclase feldspar minerals are also known to be efficient INP (Atkinson et al., 2013); however, their INA is reported to be lower than that of the K-feldspars (Zolles et al., 2015). Due to the particles in the examined dusts being complex mixtures of several types of minerals, it is not possible to disentangle the effects of potassium and plagioclase feldspar minerals on the observed INA. Boose et al. (2016a) reported that in the temperature range of 238−245 K, which is similar to the temperature range investigated here, the total content of feldspar and quartz in polydisperse aerosol is more important in determining the INA of mineral dust than the feldspar content alone. In the study presented here the highest total content of quartz and feldspar is present in the China dust (> 53%), the second most IN-active; however, this total content for the Himalaya dust is not much less at over 48%. Schütz and Sebert (1987) have reported that quartz is found mostly in the coarse mode aerosol ( > 1 μm), meaning that the large amounts of bulk quartz visible in Table 1 are likely non-representative of the mineralogy of 200 and 400 nm dust particles, rendering the effect of quartz on the observed INA insignificant. For the temperature range of 233−243 K both the total feldspar and the K-feldspar content are observed to be the most important for the INA of sub-micrometre surface-collected mineral dust examined in this study. Iceland dust is the only examined dust containing the minerals of the pyroxene group (augite and enstatite). It is, however, not possible to determine whether the minerals in this group are responsible for the lowest INA among the examined dusts as augite has been reported as a fairly good ice nucleating mineral (Isono and Ikebe, 1960) and enstatite as an inefficient one (Schill et al., 2015).

## 3.2   Dust treatments

Figures 3 and 4 present the results of the dust treatments shown as $n_{s,BET}$ as a function of $RH_w$ for particle sizes of 400 and 200 nm, respectively. The temperature of 243 K is omitted due to the higher INP concentrations measured at 238 K



compared to 243 K. This allowed for a more robust statistical examination of the results, and, hereafter, the discussion focuses only on temperatures of 233 and 238 K.

### 3.2.1 Heating

As seen in Figs. 3 and 4, the INA of Iceland dust did not change as a result of heating to 573 K for two hours; this is true for both particle sizes and both temperatures shown. Figure 5 presents the results of the TGA analysis, and Table 2 shows total carbon measurements of untreated and heated dust samples. These measurements revealed that, even though Iceland dust lost the second highest amount of mass as a result of heating to 573 K for one hour (almost 4%, Fig. 5), none of this lost mass was of the organic nature (Table 2). In fact, Iceland dust contained the lowest amount of total carbon among the three dusts. This suggests that during the heating process Iceland dust lost mostly adsorbed water, and, accordingly, the INA of Iceland dust remained unchanged after heating. Corroborating the results of Conen et al. (2011), this may also suggest that the heating process did not significantly affect the IN-relevant mineralogical composition of the dusts. The effect of heating and the loss of biological proteinaceous matter and the chemical species volatile at temperatures up to 573 K on the INA of dusts is more pronounced for China and Himalaya dusts at 238 K (Figs. 3 and 4, upper middle and right panels). For both particle sizes, the INA clearly decreased following the heat treatment, with Himalaya dust exhibiting the largest absolute decrease in the INA compared to that of China dust. Combining the TGA and the total carbon measurements demonstrated that Himalaya and China dusts lost ~ 0.5% and ~ 1.5% of their mass, respectively, due to the volatilisation of proteinaceous and carbon-containing species as a result of heating. The rest of the mass lost, as seen in Fig. 5, can probably be attributed to the loss of adsorbed water. Considering what is known about the dusts and taking into account the fraction of organic, heat-sensitive species that was lost during heating, it is only possible to say that the proteinaceous and carbon-containing species present in Himalaya dust, while smaller in amount, substantially modified the INA. Similarly, while comprising roughly 1.5% of the total bulk mass, the proteinaceous and carbon-containing material of China dust was comparatively less IN-active than that of Himalaya dust. This indicates that different proteinaceous and organic species have different ice nucleation activities, and the exact speciation is required in order to determine why dusts responded differently to the heating process. The results of the heat treatment also demonstrate that it is not possible to predict the INA of surface-collected dusts at the investigated temperature and RH conditions based on the presence and amount of organic, heat-sensitive material.

The loss of active sites on the surface of the INP as a result of heating has been previously reported (Sullivan et al., 2010a; Hill et al., 2016); however, the latter study examined the INA at temperatures above 253 K, higher than those discussed here. Sullivan et al. (2010a) examined the INA of 300 nm ATD particles at 243 K and demonstrated a decrease in the INA after heating of the dust to 523 K. In the current study the decrease in the INA of dusts after heating was observed for measurements at 243 K as well (not shown); although the decrease was dust-dependent, similar to the measurements at 238 K as discussed above. When the temperature of 233 K is examined (Figs. 3 and 4, lower panels), the effect of heating on the





INA of China and Himalaya dusts disappears, in line with results previously reported by O'Sullivan et al. (2014) and Tobo et al. (2014). O'Sullivan et al. (2014) reported that heating did not affect the INA of soil dust at temperatures below 249 K, and Tobo et al. (2014) noted that at 237 K the difference in the INA of soil dust after heating was minimal. In the present study, the effect of heating disappears only as the temperature of homogeneous freezing is approached. This likely hints at a change

in mechanism from heterogeneous freezing at 238 K supported by the presence of heat-sensitive, proteinaceous compounds to homogeneous freezing in pores at 233 K as expected with a pore condensation and freezing mechanism (Marcolli, 2014). At 238 K heating decreased the number of active sites on the particle surface by as much as 1 order of magnitude (200 nm Himalaya dust in condensation freezing mode, Figu. 4, upper right panel). It is important to note here that the INA of China and Himalaya dusts at 238 K did not decrease uniformly across the examined $RH_w$ range. Figure 3 shows that in deposition

nucleation mode the decrease in INA after heating is minimal and becomes more pronounced in condensation freezing mode. Similar behaviour is found for smaller particles (Fig. 4); however, at this size the difference is also observed in deposition nucleation mode. This is contradictory to the results presented by Sullivan et al. (2010a) reporting that heating decreased the INA of ATD particles at 243 K only in deposition nucleation mode and not in condensation freezing mode. In the study presented here it is possible to conclude that IN-active proteinaceous and carbon-containing species have the largest effect on

the INA of dust at smaller particle sizes, warmer temperatures and higher relative humidities. However, the magnitude of this effect largely depends on the identity of these proteinaceous and carbon-containing species. At 233 K (Figs. 3 and 4, lower panels) heating has no significant impact on the INA of surface-collected dusts. The only exception here are the 400 nm China dust particles (Fig. 3, lower middle panel), where heating slightly increased the INA. However, this again suggests a different freezing mechanism such as pore condensation and freezing (Marcolli, 2014).

Finally, it is important to examine how the INA of different dusts compares after the heat treatment. Figure 6 presents the $n_{s,BET}$ values as a function of $RH_w$ for heated dusts only, for both particle sizes at 238 K. When compared to untreated dust results (Fig. 2, middle panels), it can be said that the heated dusts compare similarly to the untreated dusts, with Himalaya dust still being the most active even after heating. The noted difference from the untreated dusts visible in Fig. 6 is the fact that the INA of China and Iceland dusts in condensation mode converges after heating. If it is assumed that the INA of soil

dusts can be explained by the mineralogy alone, as was done in the previous section, and if it is assumed that heating of dusts to 573 K does not change the mineralogy (Conen et al., 2011), then the heating should have no effect on how dusts compare to each other with respect to their INA. Since this is not the case, the INA of untreated China dust is second highest among dusts not only because of the second highest content of total feldspar and K-feldspar, but also due to the presence of IN-active proteinaceous and organic species. The main conclusion here is that, if it is assumed that the heating of dusts to 573 K

does not change the mineralogy, then mineralogy alone cannot fully explain the observed ice nucleation behaviour of the surface-collected dusts in condensation freezing mode at 238 K, and proteinaceous and organic material on the particle surface also has a non-negligible effect on the INA of soil dust. This is in agreement with Conen et al. (2011) and Tobo et al.



(2014) who reported that for temperatures around 265 K and warmer than 237 K, respectively, the INA of soil dust is governed mostly by the content of biological and organic matter rather than the mineral phases.

### 3.2.2 Washing

Figures 3 and 4 also present the result of the IN experiments with the washed dusts. Table 3 contains results of the auxiliary
measurements of the pH and conductivity of the supernatant water after the three washing procedures (see Methodology for details). As seen in Figs. 3 and 4, washing some of the soil dusts results in an increase in its INA. After the washing procedure Himalaya dust exhibited a clear increase in its INA for all examined temperatures and sizes, with the increase present in both deposition nucleation and condensation freezing regimes. At 238 K the INA of Iceland dust also increased, albeit only in condensation freezing regime ($RH_w > 100\%$). Such behaviour has been previously hinted at by Welti (2012),
who reported that for 400 nm ATD particles washing might have resulted in a slight increase in their INA; however, the increase was deemed insignificant. The same study reported that the washing of 800 nm ATD particles resulted in a decrease in the INA. If the washing procedure is assumed to only remove the soluble material from the particle surface, there are two possible reasons for an increase in INA after washing. This could either be attributed to the low INA of removed soluble species or the exposition of active sites on the particle surface that are otherwise blocked by the soluble material.
Conductivity and pH measurements show that, naturally, most of the soluble material was removed during the first washing cycle – this is signified by the highest supernatant conductivity as well as the lowest pH levels. Similarly to the TGA, it was not possible to directly determine the chemical species removed during the washing process; however, the conductivity and pH may provide insight. Since the first washing resulted in the pH being more acidic than the third washing, the possible identity of removed species may be that of organic acids. Species such as formic acid and acetic acid, among others, have
been found in atmospheric particulate matter and in soils in many locations around the world (Sanhueza and Andreae, 1991; Khwaja, 1995; Kawamura et al., 1996; Yuan et al., 2015), and formic acid, for instance, has been reported to decrease the INA of ice nucleating macromolecules (Pummer et al., 2015). If the low weight carboxylic acids are assumed to have a low INA, their removal could, in principle, increase the INA of soil dust as seen in the results presented here. Besides soluble organics, salts may also be present on the particle surface of the soil dust, increasing the conductivity of the Milli-Q water
after washing (Table 3). Salts such as sodium chloride NaCl are also considered to constitute a fairly inefficient INP at temperatures investigated here (Kanji et al., 2017 and references therein). A size dependence of the washing effect can also be observed in Figs. 3 and 4. Washing resulted in a greater increase in the INA of larger, 400 nm Iceland and Himalaya dust particles. For 400 nm Iceland dust particles in the condensation freezing mode and for Himalaya dust particles across the whole $RH_w$ regime, washing increased the average $n_{s,BET}$ values by as much as 1 order of magnitude. Alpert et al. (2011)
have reported that soluble compounds may, in fact, block the active sites on the particle surface; however, this was only postulated for the deposition nucleation, and did not include the notion of particle size. The current study is one of the first to



report that for certain surface-collected soil dusts, washing resulted in an increase in the INA of the dusts by removing the soluble compounds of low INA and/or the exposition of the underlying active sites.

Similar to the results of the heat treatment, Figs. 3 and 4 and Table 3 reveal that the change in the INA following the washing procedure is unrelated to the amount of soluble material that was washed off the surface of the dust particles. The conductivity of supernatant water and, hence, the deduced amount of soluble material was highest for Iceland dust (Table 3), and its INA increased only in condensation freezing mode at 238 K. Based on conductivity measurements, Himalaya dust contained the lowest amount of soluble material, and, yet, its INA increased significantly across the whole RH range at both 238 and 233 K. Additionally, China dust, which is the dust containing the second highest amount of soluble material (Table 3), did not respond to the washing procedure and its INA remained unchanged at all temperatures and particle sizes (Figs. 3 and 4). This again indicates that different soluble compounds have different ice nucleation activities, and that it is not possible to predict the INA of surface-collected dusts at the investigated temperature and RH conditions based on the presence and the amount of soluble compounds.

Figure 7 presents $n_{s,BET}$ as a function of $RH_w$ for washed dusts only, for both particle sizes at 238 K. Similar to Fig. 2 (middle panels), Fig. 7 shows that the INA of Himalaya dust after washing remains the highest among the three dusts. In the previous section about the heated dusts it was reported that the INA of China and Iceland dusts in condensation mode converged after heating, and the same can be said about the washed dusts. China dust, as noted above, did not respond to the washing procedure and maintained its original untreated INA (Figs. 3 and 4, middle panels). The INA of Iceland dust increased in condensation mode only, to $n_{s,BET}$ values similar to that of China dust. This argument again demonstrates that mineralogy alone is not able to explain the observed INA of the untreated dusts because washing, presumably, does not affect the mineralogy, and, yet, the INA of Iceland dust in condensation mode increased significantly enough to be comparable to that of a dust more active in the untreated state. Therefore, the INA of the untreated Iceland dust may be the lowest not only due to the lowest amounts of total feldspar and K-feldspar minerals, but also due to a significant presence of relatively IN-inactive soluble material blocking the active sites on the particle mineral surface. The amount and the exact chemical species of the soluble material on the surface of the soil dust particle, as well as the morphology and mineralogy of the underlying surface most likely exert a non-trivial influence on the INA of the soil dust, and the investigation of these characteristics would likely enhance the understanding of how soluble compounds on the surface of an INP affect its INA. This can be especially true in the realm of cloud processing, during which the soluble material may be redistributed/removed completely from a particle surface as a result of previous CCN/INP activation.

### 3.2.3  $H_2O_2$ treatment

Of the three treatments applied to the surface-collected soil dusts in this study, the treatment with $H_2O_2$ provided fairly inconclusive results. As seen in Figs. 3 and 4, for Iceland and China dusts the treatment with $H_2O_2$ did not affect their INA at





any temperatures or sizes investigated, while the INA of Himalaya dust increased after the $H_2O_2$ treatment. Considering similar $H_2O_2$-treated soil dust measurements and previously published results (Conen et al., 2011; O'Sullivan et al., 2014; Tobo et al., 2014; Hill et al., 2016), the initial hypothesis was that the $H_2O_2$ treatment would reduce the INA of dusts. As it turned out, the initial hypothesis had to be rejected as the $H_2O_2$ treatment either resulted in an increase in the INA or did not

affect it at all. Before an attempt is made to decipher the reasons behind the observed unexpected behaviour, it may be worthwhile to examine in detail the procedure of the $H_2O_2$ treatment. As mentioned in the methodology section, the addition of $H_2O_2$ to the dust resulted in bubbling and in a significant heat release, heating up the mixture. Once the bubbling subsided, the slurry was still heated to 323–333 K. Thus, it is not possible to completely separate the effects of heating and $H_2O_2$ on the dusts' surface properties. Certainly, both heating and $H_2O_2$ are aimed at removing a fraction or all of the organic matter from

the particle surface. However, if one is to assume that heating or $H_2O_2$ may also somehow influence the inorganic fraction, then the effects of both heating and $H_2O_2$ treatments become more complex and difficult to disentangle with respect to their effects on the INA of soil dust. Once the bubbling of the slurries completely stopped, the mixtures were stored overnight. Since $H_2O_2$ is thermodynamically unstable and decomposes to form $O_2$ and $H_2O$, the dusts were also exposed to water for a significant amount of time. Furthermore, the slurries were then diluted with deionised water by a factor of 100 before

separation. All this suggests that besides the effect of $H_2O_2$ on the chemistry of the soil dust particles, an effect of water akin to that of the washing procedure is also to be expected. In general this indicates that the effect of $H_2O_2$ treatment also includes, to an extent, the effects of heating and washing of the soil dust, and that, given the treatment procedure, it is not possible to exclusively separate the effect of $H_2O_2$ from that of washing and heating. A change in the INA after the $H_2O_2$ treatment (e.g. Himalaya dust) may be somehow explained by the difference in the INA of the removed and exposed particle

surface species as was previously seen after the washing of the Iceland dust, i.e. the removal of certain species from the particle surface frees up the active sites. At the same time, the fact that $H_2O_2$ treatment did not have any visible effect on the INA of Iceland and China dust is puzzling. Assuming no measurement, instrumental, numerical and interpretation errors, the overall INA of the species that were removed by $H_2O_2$, medium heat and some washing has to be similar to that of whatever surface properties were exposed after the removal of said species. Additionally, Mikutta et al. (2005) provided a

comprehensive overview of the effect of $H_2O_2$ on mineralogy itself and reported that, contrary to the initial assumption that $H_2O_2$ only affects the organic fraction, mineral phases can be altered by $H_2O_2$ as well. The study showed that $H_2O_2$ can result in the disintegration of the expandable clay minerals, transformation of vermiculite into mica-like structures and dissolution of poorly crystalline minerals. Since all three dusts contain minerals of the clay group and Iceland dust contains some vermiculite, changes to mineralogy induced by the $H_2O_2$ treatment cannot be ruled out. These potential changes cannot be

confirmed or quantified, and their effect on the INA of dusts remains unknown, adding yet another level of uncertainty related to the IN results of the $H_2O_2$-treated dusts.



Other uncertainties related to the changes to the particle surface induced by treatments are addressed in the following section. How the combination of the assumed effects of the $H_2O_2$ treatment and the resulting INA of dusts are related to the question of what on the particle surface is important for the ice nucleation and what is not is discussed in the conclusion section.

## 3.3     Surface area uncertainties and limitations

In the preceding sections the discussion about the soil dusts and their INA revolved mainly around the $n_{s,BET}$ values, which were calculated using the BET surface area (Table 4). The table shows that the surface area of the dust particles is rather small, 1–2 orders of magnitude lower than, for example, for porous illite NX particles (Broadley et al., 2012; Hiranuma et al., 2015), although similar to the laboratory-generated hematite particles (Hiranuma et al., 2014). When considering the surface area of untreated and treated dusts, the initial hypothesis was that untreated particles would have smaller surface area, as the cracks, steps and cavities that are commonly present in dust particles (Pruppacher and Klett, 1997) would be blocked and/or covered by organic and soluble compounds. This would possibly explain the low surface area of untreated dust particles. However, as seen in Table 4, the hypothesis did not stand, and in most cases the surface area of treated particles was either lower after the treatment or remained similar. This means that after the organic and soluble compounds were removed and the cavities, cracks and steps were presumably exposed, the surface area actually decreased, which is somewhat counter-intuitive. Such a decrease could be expected if, for instance, the untreated dust particle would be of very irregular shape, more irregular than the underlying mineral morphology. Figure 8 shows three scanning electron microscopy images representative of each of the three untreated dusts. The images show fairly normal surface morphologies expected from mineral dust particles, with cracks and cavities easily visible. Of the three treatments, the only treatment with consistent results is the washing procedure (Table 4), as the surface area of all three dusts decreased after washing. After heating the surface area also decreased with the exception of Himalaya dust, for which the surface area slightly increased. The notion that has not been addressed thus far and that is, at this point, impossible to accurately assess is the potential mechanical damage to the dust particles during the treatments. As mentioned previously, it is expected that none of the treatments are supposed to alter the mineralogy of the particles; however, it seems as though the possibility of changes to the mineral morphology cannot be excluded. For example, the washing procedure was the most rigorous one (see Methodology, Sect. 2.1). It is possible that such harsh treatment led to the collapse of the cracks, cavities and other irregularities. Some irregular fragments present on the particle surface could have also completely broken off the parent particle. Such changes to the particle morphology could potentially explain a decrease in particle surface area after the washing; however, this same argument cannot be applied to the results of the heated dusts, as 573 K heat is supposed to be rather mild and should not affect the mineral morphology (Conen et al., 2011).




Another limitation is that the BET surface area determination is conducted in bulk, similar to many other auxiliary measurements presented here, making it difficult to say how representative the bulk BET surface area is of the 200 and 400 nm particles. Inhomogeneities in the sample are possible, and the examination of all available SEM images reveals a wide variety of particle morphologies across sub-micrometre sizes. Moreover, as explained in the methodology section, the BET

measurement itself was preceded by the evacuation of the sample at 353 K heat for 15 hours. Returning to Fig. 5, one can see that a non-negligible amount of particle bulk mass was lost already at 353 K. Of the total amount of mass lost after heating the particles for one hour at 573 K, a third was lost already at 353 K. As discussed previously, this loss can quite likely be attributed to the loss of adsorbed water. However, even if it is assumed that bulk BET surface area is applicable to the 200 and 400 nm particles, the untreated BET values reported in Table 4 are not, strictly speaking, representative of the truly

untreated particles as changes to the particle surface morphology cannot be ruled out. This also applies, possibly to a lesser degree, to the BET measurements of the treated dusts.

Since $n_s$ is sensitive to the surface area, the uncertainties of the BET surface area and its measurement discussed above question the appropriateness of the $n_{s,BET}$ parameterisation. Therefore, the $n_{s,BET}$ values that form the basis of the discussion of the INA of dusts in this paper are as uncertain as the BET surface area itself. It is important to keep this notion of

uncertainty in mind when discussing the INA of atmospheric mineral dust in terms of $n_{s,BET}$. One of the reasons for the widespread use of the $n_s$ parameterisation in the IN community is that it removes the effect of particle size when examining the INA of any given species. The initial basic assumption of using $n_s$ is chemical uniformity across particles sizes. This assumption is likely not met in this study as for any given untreated dust, the 200 nm particles have higher $n_{s,BET}$ values than the 400 nm particles, indicating a varying chemical composition across sizes. The varying mineralogical composition across

sizes is, in fact, a well-known phenomenon (e.g. Atkinson et al., 2013; Boose et al., 2016a), calling into question the applicability of the $n_s$ parameterisation when examining the INA of species with non-uniform composition as a function of size. Activated fraction AF, though dependent on the size, is another parameter that is frequently used to describe the INA of any given species. In Sect. 3.1 it was shown that if the INA of untreated dusts is compared based on AF values, the conclusion is different than if the comparison is based on the $n_{s,BET}$ values, with China dust exhibiting highest AF values

across the whole examined $RH_w$ spectrum (Fig. 1). The main reason for this difference is the fact that the BET surface area of Himalaya dust is 1 order of magnitude smaller than that of the other dusts (Table 4), having an immediate and significant impact on how the INA of a selected dust is perceived. Had the discussion of the INA of dusts been based on the AF values instead of $n_{s,BET}$ values, some of the arguments and conclusions, especially, for example, the effect of mineralogy on the INA of soil dust, would have to be reconsidered.

Some of the questions that remain open in this discussion are related to whether the $n_s$ parameterisation is an appropriate quantity for examining the INA since it depends so much on the, presumably uncertain, surface area of a particle. Additionally, the points to be addressed in the future include how the INA expressed as $n_s$ compares to that based on AF and





why there are differences, and which of these two frequently used parameters is truly representative of the INA of a species in the atmosphere.

## 4    Conclusion

The INA of untreated dust particles correlated well with the total feldspar and K-feldspar content, corroborating previously

published results. It was shown that at temperatures of 238–243 K, the ice nucleation activity of the untreated, surface-collected soil dust in condensation freezing mode can be roughly approximated by one of the existing surrogates for the atmospheric mineral dust, such as illite NX, for example. The results of the heating and washing treatments revealed that mineralogy alone is not able to fully explain the observed INA of surface-collected dusts at the examined temperature and relative humidity conditions. It was shown that all untreated dusts contained a certain amount of soluble material and some

also contained heat-sensitive, organic material; however, the dusts responded differently and inconsistently to each treatment. The magnitude of change in the INA of treated dusts was not proportional to the amount of the removed heat-sensitive, organic or soluble compounds. In order to explain the observed ice nucleation behaviour it was necessary to attribute the loss or gain of dusts' INA to the INA of particular species, the exact identities of which are unknown. Within the current context, the variability in the INA of individual soluble and/or heat-sensitive, organic species is only an

assumption, and, considering that the soil dust particles are complex mixtures of various compounds, their examination is beyond the scope of this work. Part of the interpretation is also complicated by the bulk nature of treatments and auxiliary measurements and the single-particle nature of the IN experiments. Within the framework of this study it is, therefore, not possible to predict the INA of surface-collected soil dust based on the presence and amount of certain minerals or any particular class of compounds, such as soluble or proteinaceous/organic. Due to the complex nature of atmospheric mineral

dust, it is postulated that a fraction of mineral dust will always nucleate ice, regardless of its exact origin and composition. The validity of this hypothesis has been probed in several previously published studies by conducting ambient measurements of ice crystal residuals and interstitial aerosol particles in various cloud types. Knopf et al. (2014) reported that particles in the atmosphere nucleating ice seem to not be of any particular special composition or predisposition to ice nucleation. Additionally, studies on several INP types in the immersion mode have shown that ice nucleation in this mode can be

accurately predicted by temperature and solution water activity alone, with little to no dependence on the exact nature of an INP (Knopf and Alpert, 2013). Alternatively, Schmidt et al. (2017) demonstrated that the chemistry of INP and interstitial aerosol particles is, indeed, different under the mixed-phase cloud regime at a high Alpine station Jungfraujoch. The question of whether and to which extent the chemical composition of atmospheric aerosol particles, namely mineral dust, affects the likelihood of an aerosol particle to act as an INP remains open. More ambient studies examining the chemical composition of

ice crystal residuals and interstitial aerosol under various cloud regimes are required to properly assess the significance of the chemical composition of mineral and soil dust in the atmospheric ice crystal formation.



**Competing interests**

The authors declare that they have no conflict of interest.

**Acknowledgements**

MP has received funding from the European Union's Horizon 2020 research and innovation programme under the Marie
Sklodowska Curie grant agreement No 751470 "ATM-METFIN". ROD acknowledges funding from SNF grant number
200021_156581. Dr. Pavla Dagsson Waldhauserova, Dr. Luisa Ickes and Dr. Federico Bianchi are gratefully acknowledged
for the collection and provision of the surface-collected dusts examined in this study. The authors would like to thank Dr.
Eszter Barthazy and ScopeM for conducting the SEM and EDX analyses, Dr. Joanna C.H. Wong from the Laboratory of
Composite Materials and Adaptive Structures for her training and help with the TGA, Dr. Michael Plötze from the Institute
for Geotechnical Engineering for training and help with the BET surface area measurements, and Kurt Barmettler from the
Institute of Biogeochemistry and Pollutant Dynamics for assistance with the XRD and total carbon measurements. Prof.
Martin H Schroth, Prof. Kristopher McNeill and Dr. Nadine Borduas from the Institute of Biogeochemistry and Pollutant
Dynamics are all acknowledged for their help with additional measurements and equipment. MP and ZAK would also like to
sincerely thank Prof. Ulrike Lohmann and Dr. James Atkinson for plenty of helpful discussions and Fabian Mahrt for help in
the laboratory.

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




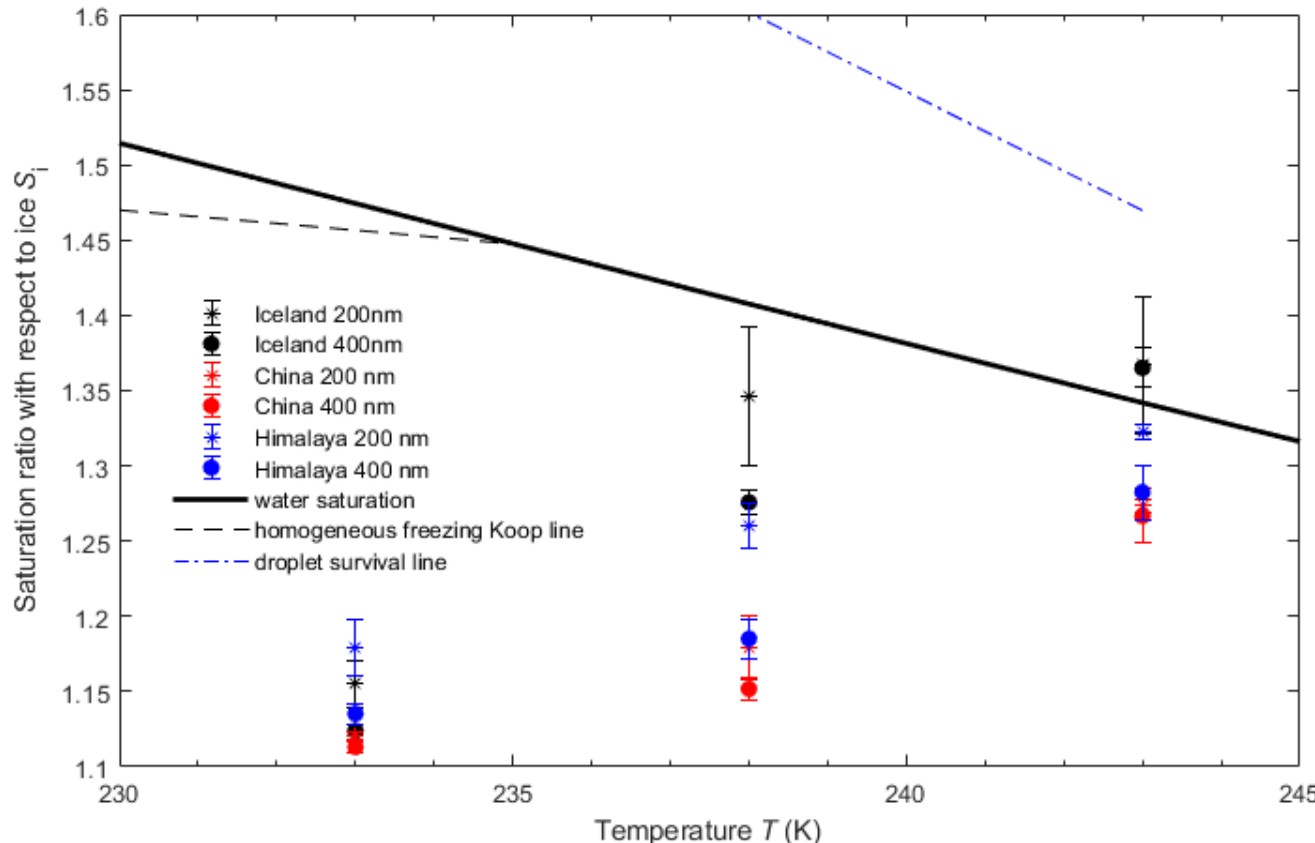

**Figure 1: Average ice nucleation onset conditions (AF = 0.1%) for each untreated dust and size. Error bars represent 1 standard deviation. Also shown are the water saturation line (RH$_w$ = 100%), the homogeneous freezing of solution droplets line (Koop et al., 2000) and the droplet survival line for PINC.**





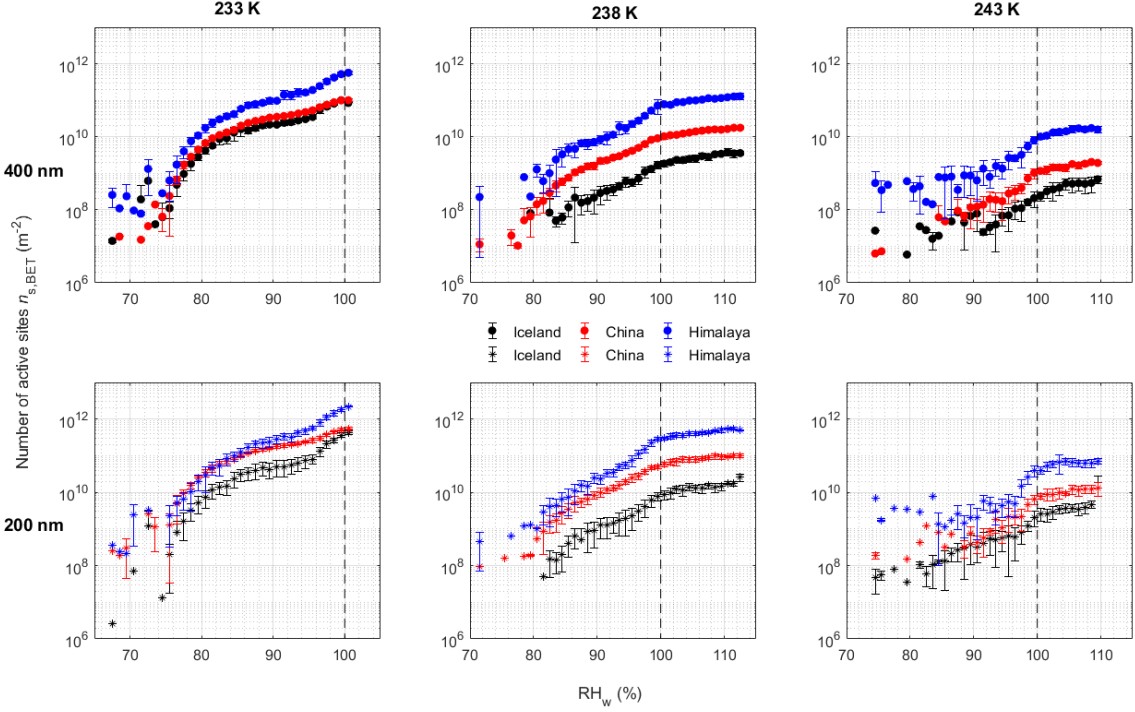

**Figure 2: Average $n_{s,BET}$ as a function of RH$_w$ for each untreated dust, size and temperature. The values are averaged into 1% RH$_w$ bins, and the error bars represent 1 standard deviation. The vertical dashed line is the water saturation (RH$_w$ = 100%) line.**



**Table 1: Mineralogical composition of each untreated dust expressed in percent weight. Included in the table are various important mineralogical groups, calculated as sums of the corresponding minerals.**

|  |  | Iceland | China | Himalaya |
|---|---|---|---|---|
| Quartz |  | 1.2 | 26.7 | 14.8 |
| Feldspar |  | 13.9 | 26.7 | 33.6 |
|  | K-feldspar | 1.9 | 6.6 | 9.3 |
|  | Plagioclase feldspar | 12.0 | 20.1 | 24.4 |
| Clay minerals |  | 5.0 | 17.0 | 9.1 |
|  | Muscovite/Illite |  | 11.8 | 6.6 |
|  | Vermiculite | 2.2 |  |  |
|  | Chamosite 1MIIb |  | 3.2 | 2.5 |
|  | Kaolinite | 2.8 | 2.1 |  |
| Amphiboles |  |  | 1.0 |  |
| Pyroxenes |  | 8.8 |  |  |
| Zeolite group |  | 5.6 |  | 0.3 |
| Ilmenites |  | 0.6 |  |  |
| Amorphous |  | 64.8 | 28.6 | 42.2 |



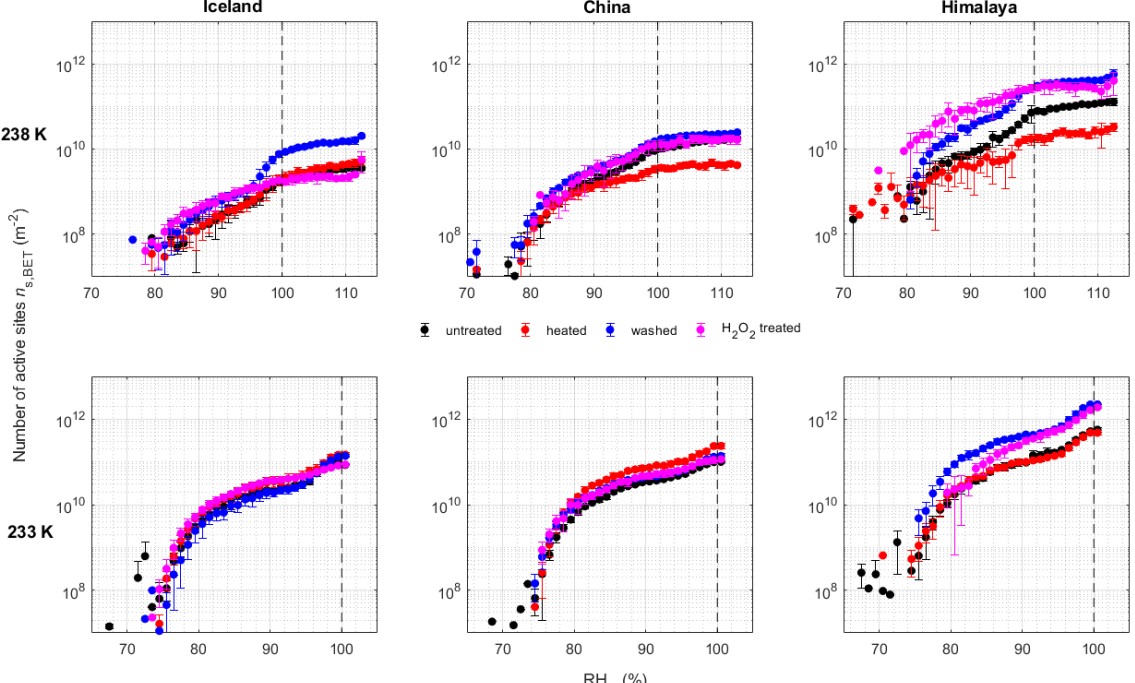

**Figure 3: Average $n_{s,BET}$ values as a function of $RH_w$ for the untreated and treated dusts for 400 nm particles. Results for temperatures of 238 and 233 K are shown. The values are averaged into 1% $RH_w$ bins, and the error bars represent 1 standard deviation. The vertical dashed line is the water saturation ($RH_w$ = 100%) line.**



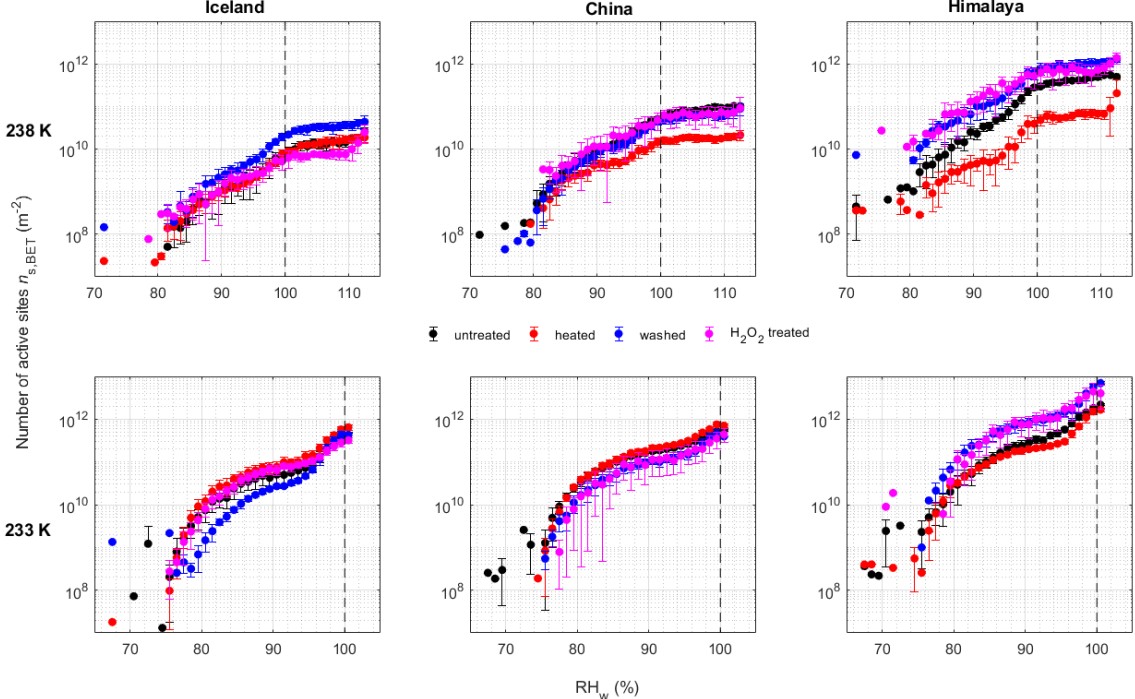

**Figure 4: Average $n_{s,BET}$ values as a function of $RH_w$ for the untreated and treated dusts for 200 nm particles. Results for temperatures of 238 and 233 K are shown. The values are averaged into 1% $RH_w$ bins, and the error bars represent 1 standard deviation. The vertical dashed line is the water saturation ($RH_w = 100\%$) line.**





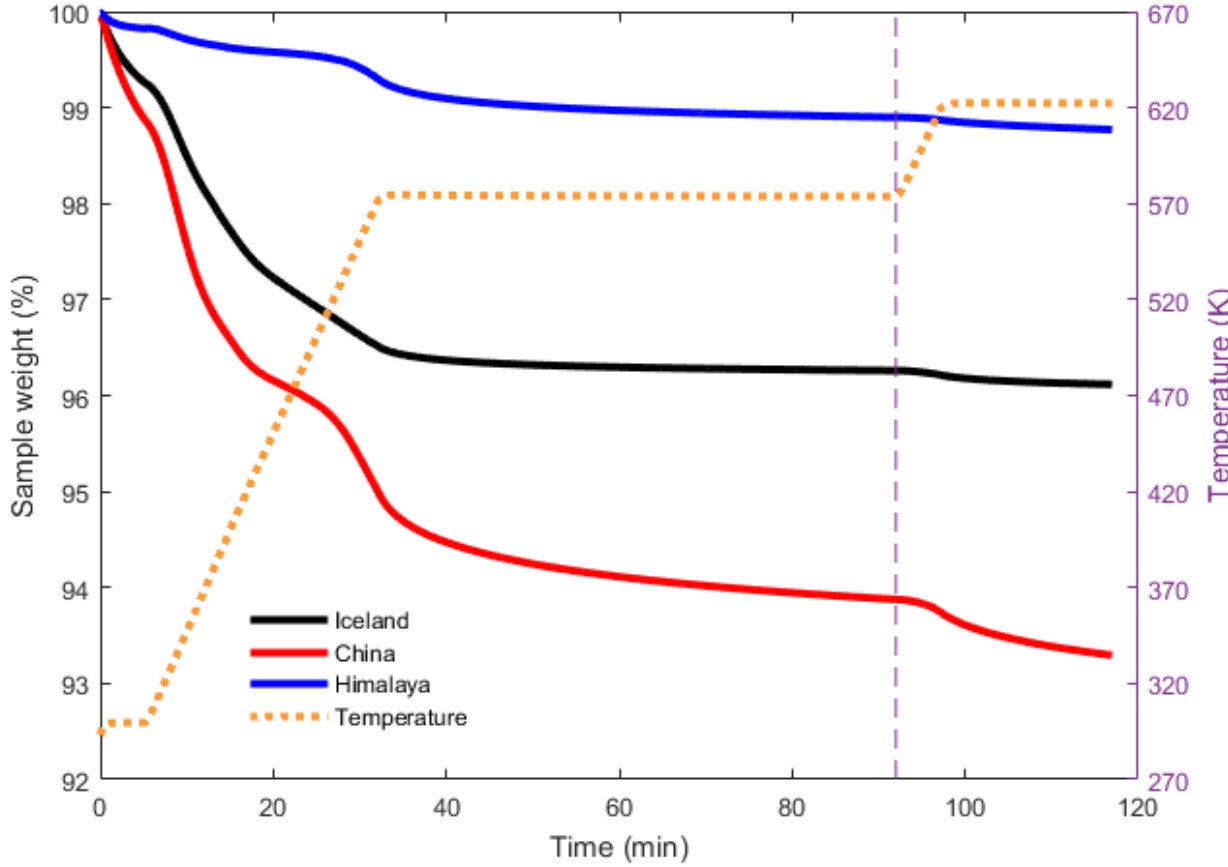

**Figure 5: Results of the thermogravimetric analysis (TGA) for the three untreated dusts, showing the temporal evolution of the mass of the untreated dust samples and the temperature. For each dust the curve is an average of three measurements. The vertical grey dashed line indicates the time when the dust had been exposed to 573 K for one hour and when the temperature started to increase further up to 623 K.**





**Table 2: Total carbon (%C) of the untreated and heated dust samples. Shown is the average value of the two measurements and 1 standard deviation.**

|              | untreated         | heated            |
|--------------|-------------------|-------------------|
| **Iceland**  | $0.185 \pm 0.004$ | $0.187 \pm 0.004$ |
| **China**    | $2.305 \pm 0.079$ | $0.985 \pm 0.004$ |
| **Himalaya** | $0.716 \pm 0.011$ | $0.200 \pm 0.008$ |



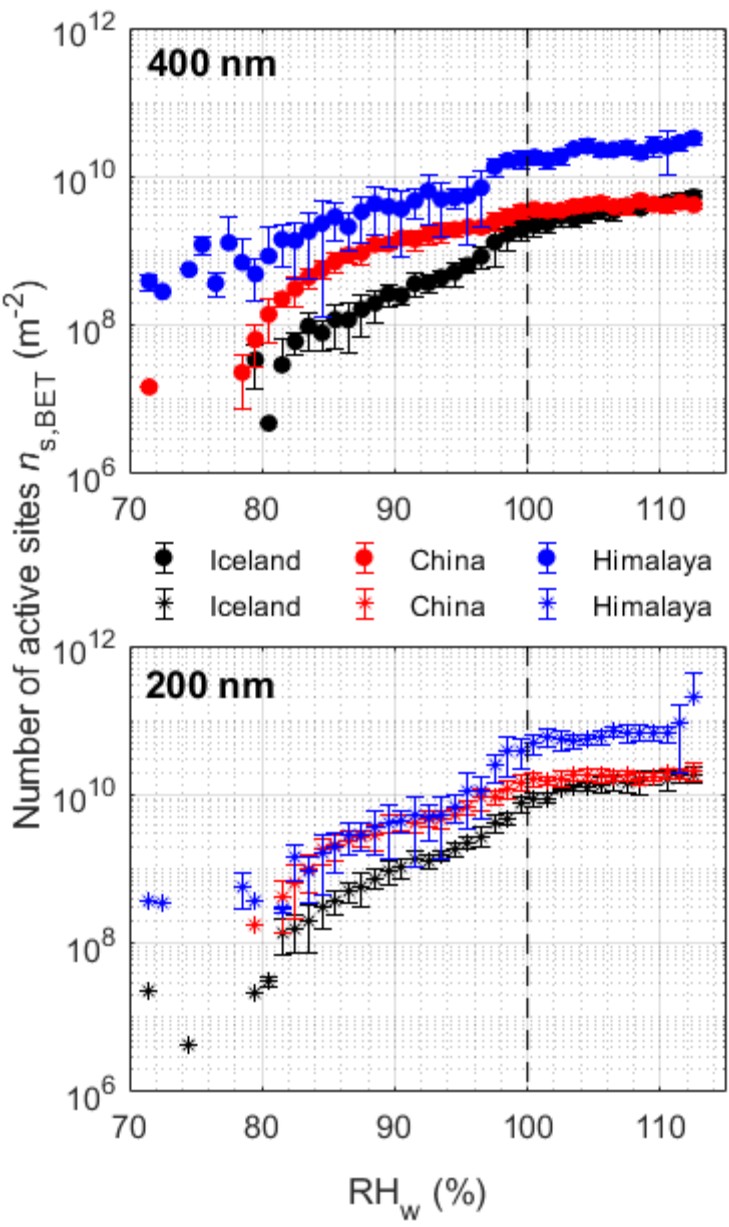

**Figure 6:** Average $n_{s,BET}$ values as a function of $RH_w$ for three heated dusts of 400 and 200 nm in diameter at 238 K. The values are averaged into 1% $RH_w$ bins, and the error bars represent 1 standard deviation. The vertical dashed line is the water saturation ($RH_w = 100\%$) line.





**Table 3: Conductivity and pH of the supernatant water of the three washing procedures of the untreated dusts. The conductivity of pure Milli-Q water was 2 μS cm$^{-1}$.**

|  | Iceland | China | Himalaya |
|---|---|---|---|
| **1st washing** |  |  |  |
| conductivity (μS cm$^{-1}$) | 128.0 | 55.3 | 14.6 |
| pH | 7.1 | 7.2 | 7.3 |
|  |  |  |  |
| **2nd washing** |  |  |  |
| conductivity (μS cm$^{-1}$) | 35.9 | 36.5 | 11.3 |
| pH | 7.6 | 7.7 | 7.3 |
|  |  |  |  |
| **3rd washing** |  |  |  |
| conductivity (μS cm$^{-1}$) | 26.3 | 26.1 | 9.2 |
| pH | 7.9 | 7.9 | 7.4 |



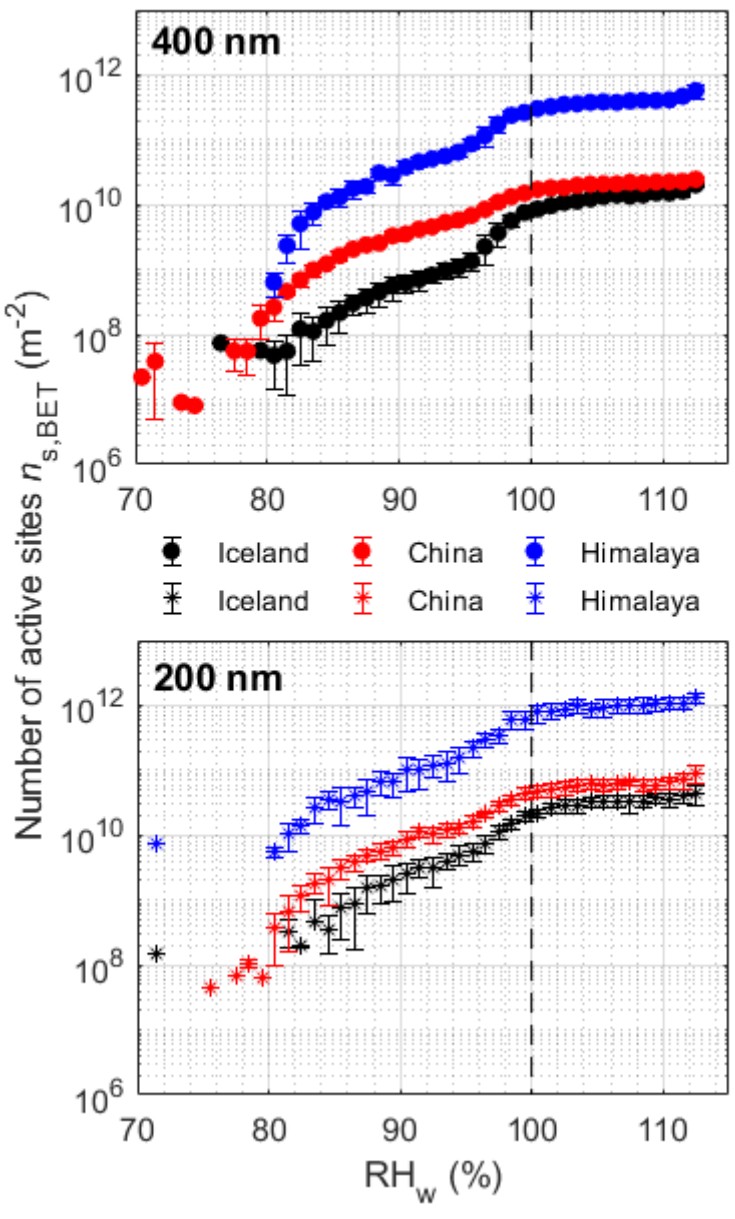

**Figure 7: Average $n_{s,BET}$ values as a function of $RH_w$ for three washed dusts of 400 and 200 nm in diameter at 238 K. The values are averaged into 1% $RH_w$ bins, and the error bars represent 1 standard deviation. The vertical dashed line is the water saturation ($RH_w$ = 100%) line.**



**Table 4: BET surface area (m$^2$ g$^{-1}$) of all untreated and treated dusts.**

|                    | Iceland | China | Himalaya |
|--------------------|---------|-------|----------|
| **untreated**      | 12.7    | 17.0  | 1.4      |
| **heated**         | 9.2     | 9.7   | 1.5      |
| **washed**         | 10.7    | 14.9  | 0.6      |
| **H$_2$O$_2$-treated** | 14.0 | 17.1  | 0.7      |





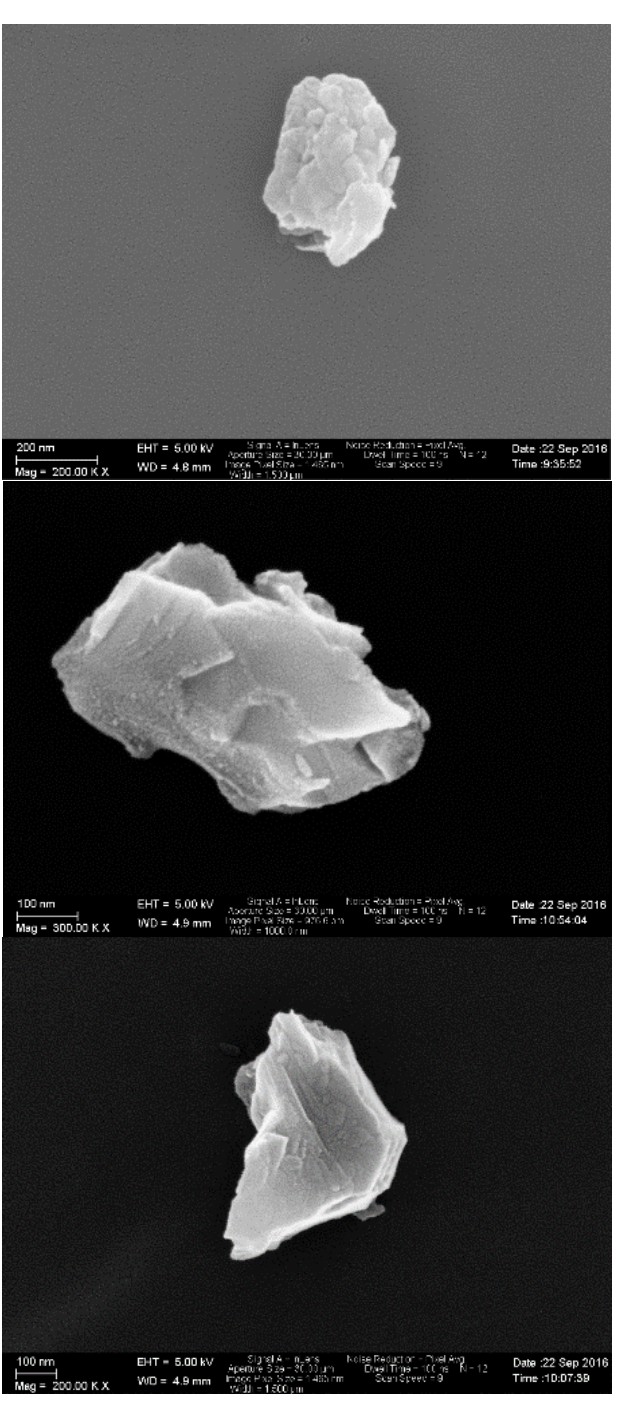

5  **Figure 8: Scanning electron microscopy (SEM) images of the untreated Iceland (top), China (middle) and Himalaya (bottom) dust particles. Of all images taken, the images presented here are the most representative of morphology and size.**