# Peer review of "A laboratory investigation of the ice nucleation efficiency of three types of mineral and soil dust"

_Atmospheric Chemistry and Physics, 2018_

## Referee Comment (RC1) · Anonymous Referee #1 · 20 Aug 2018

This paper provides a thorough investigation of the different IN active components on mineral and soil dust by using various treatments on the dust and then testing the INA pre and post treatment. It is a very nice paper that is well written and adds to the body of literature on soil dusts and their INA.

Major Comments: Page 5 line 14: More information is needed about the soil collection. What was used to collect it? How was it transported and stored? How many cm below the surface was collected? Was it all from one spot or was random sampling done over a gridded area and the samples combined to give a representative sample? Please give information about the specific sieve you used and the procedure. Was it sieved by

hand or with a machine?

I very much enjoyed the discussion of the surface area uncertainties and how the washing and heating procedures can alter the surface area. However, it makes me wonder if the paper would benefit from also including ns,geo as a means to compare to other studies that use ns,geo and to see if the results hold (meaning Himalayan ns is highest, Iceland lowest etc.). It might be an interesting exercise, especially since you call out BET as potentially not being a good way to go about calculating ns. Adding ns,geo to the discussion would complete the thought exercise of which is the best way to present INA data.

Minor Comments: Page 5 lines 7- 12: It would be nice to include the latitude and longitude of these locations.

Page 9 lines 23-24: Instead of "The study" it would be clearer to combine with the previous sentence to read "...as presented in Ullrich et al. (2017), which investigated particles in polydisperse...". Or if you think that's too long of a sentence revise to say "The Ullrich et al study investigated..."

Page 9 lines 17-25: It is not necessary, but would aid the reader to have the Kanji et al 2011 and Ullrich et al. 2017 data on a plot with your data (maybe in the supplemental) to visualize the comparison you are describing here.

Page 10 5-7: This sentence is repetitive.

Page 10 lines 22-26: "While it is not possible to directly determine the reasons behind this observed difference, it may be possible that large particles contain more soluble material blocking the active sites and/or that small particles may contain more IN-active material on their surface, e.g. bacteria or active minerals. Particles of approximately 200 nm in size, including mineral dust species, have previously been reported as constituting the majority of the INP found in the ice crystal residual size distributions (Mertes et al., 2007)." Is the activated fraction higher for the 200 nm particles as well?

Page 14 line 8: Figu. 4 change to Fig. 4

Page 14 lines 9-10: "Figure 3 shows that in deposition nucleation mode the decrease in INA after heating is minimal and becomes more pronounced in condensation freezing mode" change to "Figure 3 shows that in the deposition nucleation mode the decrease in INA after heating is minimal and becomes more pronounced in the condensation freezing mode".

Page 14: Throughout this discussion "the" should be added before the different ice nucleation modes. Example "the condensation mode".

Page 16: "The conductivity of supernatant water and, hence, the deduced amount of soluble material was highest for Iceland dust" Would different molecules or types (ie organic acids versus salts) affect the conductivity differently? Can you say 1:1 that the magnitude of the increase relates to the amount of soluble material? Or would salts increase the conductivity more than organic acids? I'm asking just out of pure ignorance to this type of measurement. It may help other readers who are not familiar if you add a sentence about this whether it would or wouldn't change the conductivity.

Page 17: I really enjoyed the detailed discussion of the H2O2 procedure and how each step may impact the results and thus the implications. It was well thought out and clearly explained so the reader could follow along with the logic. Nicely done.

Figure 8: I know it is written out in the figure caption, but it would be easier for the reader if the SEM images were labeled right on the image with Iceland, China, and Himalaya. Maybe on the upper right hand corner of each image. It would just be easier to compare them without having to remember the order.

Page 20: "It was shown that at temperatures of 238–243 K, the ice nucleation activity of the untreated, surface collected soil dust in condensation freezing mode can be roughly approximated by one of the existing surrogates for the atmospheric mineral dust, such as illite NX, for example."

If this is true, then why does one even bother with the post-treatments? Also, I would not expect a 1:1 correlation for the amount of material removed and the decrease or increase in INA because it is not linear. It is based on active sites but those may not be evenly distributed on the particles and coatings covering active sites can be covering the whole particles or a blob on one side and so the mass of something removed will not always cover the same surface area. This is especially true when you are looking at bulk removal across all sizes and then trying to correlate that to size selected particles. I would not expect it to be linear or to have a simple relationship. This ties back to what you were saying about bulk measurements complicating matters and dust being known to have varying chemical composition with size. The discussion might benefit from a little bit more explanation about these complexities.

---

## Referee Comment (RC2) · Anonymous Referee #2 · 10 Sep 2018

In this study the authors carry deposition and condensation freezing experiments to study the ice nucleation activity of different soil samples collected around the world. The authors study the effect of size and temperature as well as of different treatments applied to the particles. It is found that the active site density correlates well with the Feldspar content of the samples pointing to a mineralogy control of the ice nucleation activity. However the effect of different heating, washing and chemical treatment is not consistent with such a hypothesis. This is an interesting and detailed study of ice nucleation of dust samples relevant for cloud formation. The authors emphasize the limitations of the active site density approach and warn about the assumption of the mineralogy control of the ice nucleation activity of dust samples. The paper is well

written and organized. I recommend its publication in ACP after some comments are addressed.

My only general comment deals with the slight but important differences between immersion and condensation freezing. It has been shown in cold stage studies of immersion freezing that slight modifications in the environment around the droplet have large effects on the measured active site density (like for example a droplet evaporating during the experiments). Condensation freezing is by definition a non-equilibrium measurement since the droplets are presumably changing their size during activation. Even for deposition it is likely the water coverage of the particles changes during the ice nucleation measurements. So my question is: how does the water adsorption and eventual droplet activation process affect the ice nucleation measurements? For example by virtue of the Kelvin effect when exposed to a given RHw the 400 nm particles would activate more easily and tend to absorb water more rapidly than the 200 nm particles. This could explain why they apparently look more active, when in reality they may have been exposed to a different thermodynamic environment. Washing may remove some of the soluble material hence the particles may condense and activate more easily, thus leading to an effect on cloud condensation nuclei activity that could be mistaken by an effect on ice nucleation. The authors should comment on how this may affect their measurements.

Technical comments: Page 2, Line 17. Must be "higher" instead of "warmer" temperature.

Page 5, Line 4. Deposition defined as RHw<100% seems ambiguous. I know this is standard practice, but a can dust particle adsorb several monolayers of water at RHw<100 %? How long does it take for a dust particle to reach equilibrium coverage when exposed to a given RHw? Is the residence time of the instrument long enough for it to happen?

Page 8, Eq. 1. Please specify whether this is the surface area per particle or the total

surface area in the population.

Page 9, Line 6. Is the increase in AF consistent with CNT predictions? Although CNT predicts an increase in AF with area it tends to be much more subtle than typically measured.

Page 9, Line 10. This is unexpected since dust is assumed to be a good ice nucleating particle. Is the fact that the maximum AF reaches only 1% due to residence time or particle size? Would it be the higher/lower in the atmosphere?

Page 13, Line 8. Recent work has shown that the dust surface morphology may be more delicate that previously thought. For example, active sites may be susceptible to the addition of very low concentration of ammonium sulfate and other solutes. Thus the heating treatment and the H2O2 hydrolysis seem harsh. Maybe for next experiments enzymatic hydrolysis could be considered to better target organic material.

Page 14, Lines 29-32 (also Page 16, Lines 19-22). The authors make this claim several times. However caution should be taken since immersion and condensation are different and in this particular case may not be completely comparable. The treatments may not only affect the ability of dust to act as ice nucleating particle but also its ability to act as a cloud condensation nuclei (CCN). See my general comment above.

Page 25, Line 23. Is the higher area of Himalayan dust a result of less aggregation or higher porosity?

Page 20, Line 7-10. This is true as long as the treatments do not affect CCN activity.

---

## Author Comment (AC1) · 5 Oct 2018

This paper provides a thorough investigation of the different IN active components on mineral and soil dust by using various treatments on the dust and then testing the INA pre and post treatment. It is a very nice paper that is well written and adds to the body of literature on soil dusts and their INA.

Response: The authors would like to thank the referee for reviewing the paper and for the praise!

Major comments

1) Page 5 line 14: More information is needed about the soil collection. What was used to collect it? How was it transported and stored? How many cm below the surface was collected? Was it all from one spot or was random sampling done over a gridded area and the samples combined to give a representative sample? Please give information about the specific sieve you used and the procedure. Was it sieved by hand or with a machine?

Response: The following sentences have been added to the main text on page 5: "All dusts were collected either right at or directly below the surface at each individual location. After collection the dusts were then stored in plastic bags in dark conditions at room temperature. To allow for ice nucleation experiments, the collected dust samples were sieved with a series of dry sieves to select only the particles below 45 $\mu$m in diameter (Retsch Vibratory Sieve Shaker AS 200)."

2) I very much enjoyed the discussion of the surface area uncertainties and how the washing and heating procedures can alter the surface area. However, it makes me wonder if the paper would benefit from also including ns,geo as a means to compare to other studies that use ns,geo and to see if the results hold (meaning Himalayan ns is highest, Iceland lowest etc.). It might be an interesting exercise, especially since you call out BET as potentially not being a good way to go about calculating ns. Adding ns,geo to the discussion would complete the thought exercise of which is the best way to present INA data.

Response: The authors completely agree with the referee comment and have, indeed, for a long time considered including ns,GEO in the manuscript. This, however, was not done for several reasons. First, ns,BET values of all untreated and treated dusts is one of the most unique aspects of the paper. As the analysis went along, it became apparent that ns,BET would become the focal point of the paper and result in the discussion related to its limitations and uncertainties. Second, since we have conducted experiments with monodisperse particles, ns,GEO values are computed directly from AF values, and they would show the same trends and results as did AF values (Fig. 1)

when it comes to comparing the INA of dusts to each other for the same size. The only clear benefit of the inclusion of ns,GEO values in the discussion would, as correctly pointed out by the referee, allow for comparison to other previously published studies. This feeds into the third reason for omitting ns,GEO values. The paper is already quite long, and the discussion of AF and ns,BET values is already rather detailed and labourious. Adding ns,GEO would require to significantly expand the manuscript further, add at least two new figures, modify almost every section of the manuscript and include ns,GEO in the discussion about which parameter is the best for describing the INA of any given species, something that has already been done by, e.g., Murray et. al. 2012 and Hiranuma et.al. 2015. I believe the significant expansion of the manuscript would not be worth it considering that the only important benefit of including ns,GEO values is the comparison to previously published studies, which, in the end, is not one of the main focal points of the paper anyway. No modifications have been made.

Minor comments

1) Page 5 lines 7- 12: It would be nice to include the latitude and longitude of these locations.

Response: The geographic coordinates have been added on page 5.

2) Page 9 lines 23-24: Instead of "The study" it would be clearer to combine with the previous sentence to read ": : :as presented in Ullrich et al. (2017), which investigated particles in polydisperse: : :". Or if you think that's too long of a sentence revise to say "The Ullrich et al study investigated: : :"

Response: For clarity, the second sentence now reads: "The mentioned study investigated. . ."

3) Page 9 lines 17-25: It is not necessary, but would aid the reader to have the Kanji et al 2011 and Ullrich et al. 2017 data on a plot with your data (maybe in the supplemental) to visualize the comparison you are describing here.

[Figure]

Response: Figure 1 modified to include data points from the two referenced studies. The text at the bottom pf page 9 and top of page 10 now reads: "Comparing the onset values to those published in a study by Kanji et al. (2011) reveals that in the 238−243 K temperature range the Iceland, China and Himalaya dusts are all more active than the Saharan (SD) and Canary Island (CID) dusts with the exception of 200 nm Iceland dust particles at 238 K (Fig. 1). At 233 K all examined dusts and particle sizes are more IN-active than the Canary Island dust (Kanji et al., 2011). The referenced study examined polydisperse particles with a mode size of 200−300 nm in diameter. The onset values presented also compare well to those of Asian desert dusts AD1 and AD2, and Saharan desert dust SD2 as presented in Ullrich et al. (2017) (Fig. 1). At the warmest temperature Saharan desert dust SD19 seems to be more ice-active than any of the dusts examined in the current study (Fig. 1). It should be kept in mind that, with the exception of dust AD2, the onset values of Ullrich et al. (2017) seen in Figure 1 were defined for AF higher than 0.001 assumed here.".

4) Page 10 5-7: This sentence is repetitive.

Response: This sentence serves as a reminder, since the discussion immediately thereafter elaborates further on the comparison of dusts and shows that which dust is most active depends on which quantity is examined (AF or ns). No modifications made.

5) Page 10 lines 22-26: "While it is not possible to directly determine the reasons behind this observed difference, it may be possible that large particles contain more soluble material blocking the active sites and/or that small particles may contain more IN-active material on their surface, e.g. bacteria or active minerals. Particles of approximately 200 nm in size, including mineral dust species, have previously been reported as constituting the majority of the INP found in the ice crystal residual size distributions (Mertes et al., 2007)." Is the activated fraction higher for the 200 nm particles as well?

Response: No, it is not. AF is higher for larger particles. This is mentioned on page

9, lines 16-17 and further addressed on page 20, lines 8-17 of the revised manuscript, where the authors pose a question as to which of the two parameters (AF or ns) is truly representative of INA of a given species.

6) Page 14 line 8: Figu. 4 change to Fig. 4

Response: Corrected.

7) Page 14 lines 9-10: "Figure 3 shows that in deposition nucleation mode the decrease in INA after heating is minimal and becomes more pronounced in condensation freezing mode" change to "Figure 3 shows that in the deposition nucleation mode the decrease in INA after heating is minimal and becomes more pronounced in the condensation freezing mode".

Response: Corrected.

8) Page 14: Throughout this discussion "the" should be added before the different ice nucleation modes. Example "the condensation mode".

Response: Definite article "the" inserted throughout the text, where appropriate.

9) Page 16: "The conductivity of supernatant water and, hence, the deduced amount of soluble material was highest for Iceland dust" Would different molecules or types (ie organic acids versus salts) affect the conductivity differently? Can you say 1:1 that the magnitude of the increase relates to the amount of soluble material? Or would salts increase the conductivity more than organic acids? I'm asking just out of pure ignorance to this type of measurement. It may help other readers who are not familiar if you add a sentence about this whether it would or wouldn't change the conductivity.

Response: Organic acids would be expected to be less conductive than salts, but this would be purely because they are weak acids and therefore partially dissociate, thus the ionic strength of the solution would be weaker than that of inorganic salts or acid solutions. Since we measure the conductivity, which should be proportional to the total ionic strength/content of the supernatant water, it is clear that the contribution of ions

to the supernatant water could come from both organics and inorganics. It could be that the water is composed of organic and inorganic ions but it would be impossible to distinguish if a lower conductivity would be due to presence of organic acids or a comparatively lower amount of inorganic ions. Similarly, we cannot speculate that a high conductivity is due to inorganic ions or a comparatively large amount of organic (acid) ions. As such, the use of the phrase "soluble material" is preferred to be inclusive of any potential contribution from both inorganic and organic compounds. Therefore, the electrolytic strength of the solution is expected to be directly proportional to the conductivity.

As such, it would serve no purpose to distinguish between the conductivity of organics vs. inorganics because the relative contribution of these species to the ionic content of the supernatant water is not known. Thus, no modifications to the revised manuscript are made in this regard.

10) Page 17: I really enjoyed the detailed discussion of the H2O2 procedure and how each step may impact the results and thus the implications. It was well thought out and clearly explained so the reader could follow along with the logic. Nicely done.

Response: Thank you!

11) Figure 8: I know it is written out in the figure caption, but it would be easier for the reader if the SEM images were labeled right on the image with Iceland, China, and Himalaya. Maybe on the upper right hand corner of each image. It would just be easier to compare them without having to remember the order.

Response: Figure 8 modified.

12) Page 20: "It was shown that at temperatures of 238–243 K, the ice nucleation activity of the untreated, surface collected soil dust in condensation freezing mode can be roughly approximated by one of the existing surrogates for the atmospheric mineral dust, such as illite NX, for example." If this is true, then why does one even bother

with the post-treatments? Also, I would not expect a 1:1 correlation for the amount of material removed and the decrease or increase in INA because it is not linear. It is based on active sites but those may not be evenly distributed on the particles and coatings covering active sites can be covering the whole particles or a blob on one side and so the mass of something removed will not always cover the same surface area. This is especially true when you are looking at bulk removal across all sizes and then trying to correlate that to size selected particles. I would not expect it to be linear or to have a simple relationship. This ties back to what you were saying about bulk measurements complicating matters and dust being known to have varying chemical composition with size. The discussion might benefit from a little bit more explanation about these complexities.

Response: The post-treatments were intended to investigate whether other, non-mineralogical compounds of the mineral and soil dust affect its INA and, as it turned out, they did. Hence, the subsequent discussion saying that mineralogy alone is not able to fully explain the observed INA. In the atmosphere, the mineral and soil dust particles are complex mixtures of various compounds, and their INA is governed by the INA of all individual species present on the particle surface. The statement mentioned by the referee serves to say that even though the dust samples examined here are complex mixtures of various compounds, their INA can still be roughly approximated by one of the existing surrogates for the atmospheric mineral dust. This demonstrates that individual compounds on the particle surface do affect its INA, but not significantly enough that they cannot be approximated by an existing surrogate.

As for the second issue raised, in several places in the manuscript it is mentioned that different individual heat-sensitive and soluble compounds must have different individual INA, and that is specifically due to the fact that the response of the dusts' INA to the removal of said species was not proportional to the amount of material removed. This is explicitly mentioned in both heating and washing sections. The point raised by the referee is also explained in detail in the washing section where not only the different

soluble species and their different individual INA may have affected the overall INA of dusts, but also the exposition of the underlying active sites that were otherwise blocked/covered by the soluble material may be important as well. The conclusion also addresses the point that the overall observed INA of dust has to be attributed to different INA of individual heat-sensitive and soluble species. The identification of individual compounds on the particles' surface and the examination of their individual INA are beyond the scope of this paper. No modifications have been made.

References:

- Hiranuma, N., Augustin-Bauditz, S., Bingemer, H., Budke, C., Curtius, J., Danielczok, A., Diehl, K., Dreischmeier, K., Ebert, M., Frank, F., Hoffmann, N., Kandler, K., Kiselev, A., Koop, T., Leisner, T., Möhler, O., Nillius, B., Peckhaus, A., Rose, D., Weinbruch, S., Wex, H., Boose, Y., DeMott, P. J., Hader, J. D., Hill, T. C. J., Kanji, Z. A., Kulkarni, G., Levin, E. J. T., McCluskey, C. S., Murakami, M., Murray, B. J., Niedermeier, D., Petters, M. D., O'Sullivan, D., Saito, A., Schill, G. P., Tajiri, T., Tolbert, M. A., Welti, A., Whale, T. F., Wright, T. P., and Yamashita, K.: A comprehensive laboratory study on the immersion freezing behavior of illite NX particles: a comparison of 17 ice nucleation measurement techniques, Atmos. Chem. Phys., 15, 2489–2518, doi:10.5194/acp-15-2489-2015, 2015.

- Murray, B. J., O'Sullivan, D., Atkinson, J. D., and Webb, M. E.: Ice nucleation by particles immersed in supercooled cloud droplets, Chem. Soc. Rev., 41, 6519–6554, 2012.

Thank you very much, again, for taking the time to read, comment and, therefore, improve the paper!
* * *

---

## Author Comment (AC2) · 5 Oct 2018

In this study the authors carry deposition and condensation freezing experiments to study the ice nucleation activity of different soil samples collected around the world. The authors study the effect of size and temperature as well as of different treatments applied to the particles. It is found that the active site density correlates well with the Feldspar content of the samples pointing to a mineralogy control of the ice nucleation activity. However the effect of different heating, washing and chemical treatment is not consistent with such a hypothesis. This is an interesting and detailed study of ice nucleation of dust samples relevant for cloud formation. The authors emphasize the

limitations of the active site density approach and warn about the assumption of the mineralogy control of the ice nucleation activity of dust samples. The paper is well written and organized. I recommend its publication in ACP after some comments are addressed.

Response: The authors would like to thank the referee for reviewing the paper and for the praise!

Comments

1) My only general comment deals with the slight but important differences between immersion and condensation freezing. It has been shown in cold stage studies of immersion freezing that slight modifications in the environment around the droplet have large effects on the measured active site density (like for example a droplet evaporating during the experiments). Condensation freezing is by definition a non-equilibrium measurement since the droplets are presumably changing their size during activation.

Response: The above is true for a cold stage study where the same particles experience an increasing or decreasing relative humidity, but in the current work, the aerosol particles are exposed to a constant RH given that their residence time in the chamber is 7 seconds and the rate of change of RHw is 2% min-1 implying that the aerosol particles see a change of $\sim$ RHw of 0.2% and as such should not be growing or changing size of the droplet significantly. This can be also confirmed from the data in Figure 2, where ns is shown as a function of RHw, and one notes that above RHw of 100%, the curve plateaus suggesting that there is no effect from having a larger droplet at higher saturation conditions, at least for the range of the supersaturated conditions studied here. However, this would be different for conditions below water saturation (see response below).

Even for deposition it is likely the water coverage of the particles changes during the ice nucleation measurements. So my question is: how does the water adsorption and eventual droplet activation process affect the ice nucleation measurements? For example by virtue of the Kelvin effect when exposed to a given RHw the 400 nm particles would activate more easily and tend to absorb water more rapidly than the 200 nm particles.

Response: If in fact "deposition nucleation" is occurring via liquid water formation, the above would be true that CCN activation could play a role, but this should only be important for conditions very close to water saturation. Let's consider two scenarios: Under the operational definition that deposition nucleation is occurring by vapour adsorbing onto the particle surface, to form a critical ice germ followed by bulk ice nucleation, the larger surface of the 400 nm particle plays a role by providing more surface for an active site to accommodate adsorption of the vapour that forms the critical ice germ. Given that our particles are largely insoluble, droplet activation is not expected until RHw is very close to or above water saturation. As such, it is not expected that eventual droplet activation would affect the ice nucleation measurements below water saturation. This is demonstrated by the plateauing of the $n_s$ curves for RHw > 100% (Fig. 2).

That being said, in a second scenario, if liquid water is being absorbed into the particles below water saturation because of the presence of soluble material, as is evident from the data in the paper (Table 3), and then this condensed water can cause freezing because of an active site present on the dust particle, then it would be the case that the increase in $n_s$ as a function of increasing RHw results from a form of hygroscopic uptake due to the soluble material, but not CCN activation, as activation into drops would only occur at supersaturated conditions. However, because of the hygroscopic uptake and resulting growth and increase in the ice active fraction (or $n_s$) would be observed as more absorbed water results on the particles with increasing RHw resulting in a more dilute hygroscopic layer.

This could explain why they apparently look more active, when in reality they may have been exposed to a different thermodynamic environment. Washing may remove some of the soluble material hence the particles may condense and activate more easily, thus leading to an effect on cloud condensation nuclei activity that could be mistaken
by an effect on ice nucleation. The authors should comment on how this may affect their measurements.

Response: If indeed liquid water is involved in the deposition regime, the above would be true, but this would be due to water absorption and hygroscopic growth rather than CCN activation. This comment is acknowledged and now clarified this in the revised manuscript on page 8, lines 11-18, page 11, lines 6-8, page 16, lines 17-19.

Technical comments

1) Page 2, Line 17. Must be "higher" instead of "warmer" temperature.

Response: Corrected.

2) Page 5, Line 4. Deposition defined as RHw<100% seems ambiguous. I know this is standard practice, but a can dust particle adsorb several monolayers of water at RHw<100 %? How long does it take for a dust particle to reach equilibrium coverage when exposed to a given RHw? Is the residence time of the instrument long enough for it to happen?

Response: It is agreed that this is just an operational definition. The adsorption and absorption of water < RHw 100% is acknowledged. See response to the main comment above.

3) Page 8, Eq. 1. Please specify whether this is the surface area per particle or the total surface area in the population.

Response: Corrected. The sentence immediately prior to the equation now reads: "The ns per particle was calculated as follows:"

4) Page 9, Line 6. Is the increase in AF consistent with CNT predictions? Although CNT predicts an increase in AF with area it tends to be much more subtle than typically measured.

Response: CNT calculations were not carried out in this study, and the reference to

[Figure]

CNT has been removed. Instead, additional references have been added, all of which invoked CNT and performed calculations. The sentence on page 9 lines 17-18 now reads "Such size dependence is consistent with previous observations (Archuleta et al., 2005; Welti et al., 2009; Kanji et al., 2011)".

5) Page 9, Line 10. This is unexpected since dust is assumed to be a good ice nucleating particle. Is the fact that the maximum AF reaches only 1% due to residence time or particle size? Would it be the higher/lower in the atmosphere?

Response: This is true and is addressed a bit more on page 10, lines 7-15 in the revised manuscript. There are several reasons why our maximum AF is ~33% (China dust, 400 nm, coldest T, highest RHw). One explanation is, indeed, the residence time in the chamber, which is nominally seven seconds. Another explanation could be the deviation of particles from the laminar sample flow, although the lab experiments showed that this deviation affects ~10% of particles (unpublished data). Yet another explanation could be that even for mineral dust, which is indeed a good INP, not all particles would activate into INP, highlighting the overall notion that INPs in the atmosphere are rare. It is very difficult to say how the AF in the atmosphere would be different from the lab experiments especially given that in the atmosphere particles are externally mixed, as such low AFs would be the norm. Additionally, lower AF would be expected in the atmosphere because the supply of supersaturated water vapour may not be continuous in the atmosphere as it is in PINC. Temperature in the atmosphere is also likely to change due to the release of latent heat of condensation.

6) Page 13, Line 8. Recent work has shown that the dust surface morphology may be more delicate that previously thought. For example, active sites may be susceptible to the addition of very low concentration of ammonium sulfate and other solutes. Thus the heating treatment and the H2O2 hydrolysis seem harsh. Maybe for next experiments enzymatic hydrolysis could be considered to better target organic material.

Response: Authors absolutely agree with and thank the reviewer for this suggestion.

7) Page 14, Lines 29-32 (also Page 16, Lines 19-22). The authors make this claim several times. However caution should be taken since immersion and condensation are different and in this particular case may not be completely comparable. The treatments may not only affect the ability of dust to act as ice nucleating particle but also its ability to act as a cloud condensation nuclei (CCN). See my general comment above.

Response: Please, see the response to the main comment and the parts of the text where additions have been made.

8) Page 25, Line 23. Is the higher area of Himalayan dust a result of less aggregation or higher porosity?

Response: The BET surface area of Himalaya dust is the smallest among all dusts (Table 4). Authors are not sure what this comment is referring to.

9) Page 20, Line 7-10. This is true as long as the treatments do not affect CCN activity.

Response: That's correct.

Thank you very much, again, for taking the time to read, comment and, therefore, improve the manuscript!

———————————————